# Kinetochore-centrosome feedback linking CENP-E and Aurora kinases controls chromosome congression

Kruno Vukušić ✉ & Iva M. Tolić ✉

Chromosome congression is crucial for accurate cell division, with key roles played by kinetochore components, the molecular motor CENP-E/kinesin-7, and Aurora B kinase. However, Aurora B kinase can both inhibit and promote congression, suggesting the presence of a larger signaling network. Our study demonstrates that centrosomes inhibit congression initiation when CENP-E is inactive by regulating the activity of kinetochore components. Depletion of centrioles via Plk4 kinase inhibition allows chromosomes near acentriolar poles to initiate congression independently of CENP-E. At centriolar poles, high Aurora A kinase enhances Aurora B activity, increasing phosphorylation of microtubule-binding proteins at kinetochores and preventing stable microtubule attachments in the absence of CENP-E. Conversely, inhibition of Aurora A or expression of a dephosphorylatable mutant of the kinetochore microtubule-binding protein Hec1 enables congression initiation without CENP-E. We propose a negative feedback mechanism involving Aurora kinases and CENP-E that regulates the timing of chromosome movement by modulating kinetochore–microtubule attachments and fibrous corona expansion, with the Aurora A activity gradient providing critical spatial cues for the network's function.

Chromosome congression, the process by which chromosomes align at the spindle equator during early mitosis, is essential for accurate chromosome segregation and cell division[1,2]. The kinetochore motor protein CENP-E (Centromere-associated protein E) is recognized as a critical player in this process[2–6]. The prevailing view of CENP-E's role during congression is that its plus-end directed motor activity moves chromosomes along microtubules to the spindle midplane, independently of their biorientation[7–10]. However, our recent observations challenge this understanding[11]. We propose that CENP-E couples chromosome congression to biorientation near the spindle poles by stabilizing end-on attachments at polar kinetochores via its interaction with BubR1[11]. Thus, chromosome congression requires conversion from lateral to end-on kinetochore–microtubule attachments[12–14]. Notably, CENP-E is essential only for the congression of chromosomes near centrosomes[8,15], and biorientation in unperturbed spindles occurs exclusively outside the centrosome-proximal regions[16,17]. Yet, the role of centrosome signaling in both chromosome biorientation and congression remains unclear.

Centrosomes are non-membranous organelles involved in various cellular processes, including cell division and ciliogenesis[18]. The majority of somatic cells in vertebrates contain two centrosomes, each centered around a pair of centrioles. Composed of microtubules arranged in a nine-fold symmetry, these cylindrical structures anchor and organize the pericentriolar material, a component crucial for microtubule nucleation and spindle formation[18]. Previous studies have demonstrated that centrioles are not essential for chromosome segregation in mitotic vertebrate cells[19,20], nor in oocytes, which naturally lack centrioles[21]. In somatic cells, centrosome-associated kinases such as Aurora A drive centrosome maturation, characterized by organelle enlargement, increased microtubule nucleation, and spindle pole establishment before mitosis[22]. Beyond its role in centrosome maturation, Aurora A collaborates with Aurora B to phosphorylate

Division of Molecular Biology, Ruđer Bošković Institute, Zagreb, Croatia. ✉e-mail: kvukusic@irb.hr; tolic@irb.hr

CENP-E, facilitating chromosome congression near centrosomes[10,23]. However, our recent findings show that Aurora A at centrosomes and Aurora B at kinetochores inhibit congression initiation in the absence of CENP-E[11]. These seemingly paradoxical activities of Aurora kinases within the same process suggest a signaling feedback loop among mitotic spindle components, the nature and regulation of which remain unknown.

In this study, we combined RNA interference, expression of phospho-mutants, acute chemical inhibition, phospho-specific antibodies, and large-scale live-cell imaging to investigate how centrosomes regulate chromosome congression under varying levels of CENP-E activity. We reveal that centrioles, through Aurora A kinase, inhibit congression of polar chromosomes when CENP-E is non-functional. Depletion of centrioles by Polo-like kinase 4 (Plk4) inhibition allowed chromosomes to initiate congression independently of CENP-E, revealing that the inhibitory effect of centrioles is mediated by high Aurora A activity at centriolar poles. Near centrosomes, Aurora A was found to overactivate Aurora B at kinetochores, which resulted in increased phosphorylation of the KMN network (Knl1 complex, Mis12 complex, and Ndc80 complex). This hyperphosphorylation prevented stable microtubule attachments and thereby blocked congression initiation. Expression of constitutively dephosphorylated Hec1 was sufficient to restore congression in the absence of CENP-E, but only after fibrous corona disassembly. We propose a negative feedback loop in which Aurora B, activated by Aurora A near centrosomes, inhibits congression initiation through two interlinked mechanisms: destabilization of end-on attachments and promotion of fibrous corona expansion. At the same time, CENP-E, activated by Aurora kinases, counteracts this destabilization to support chromosome congression. Together, these findings advance our understanding of the molecular interplay between centrosomes and kinetochores during mitosis.

## Results

### Centrioles delay the initiation of congression when CENP-E is inactive

Chromosome congression is notably delayed in cells lacking CENP-E, with polar chromosomes often lingering near the centrosomes before initiating movement toward spindle equator[8,11,24]. Based on this, we hypothesized that centrosomes might inhibit the initiation of congression of polar chromosomes when CENP-E is inactive or absent. To test this hypothesis, we imaged non-transformed human RPE-1 cells expressing CENP-A-GFP and centrin1-GFP[25] by confocal microscopy after acute reactivation of CENP-E and after depletion of CENP-E by RNA interference (Supplementary Fig. 1a, b)[11]. We tracked polar kinetochore movements over time (see Methods) and measured the duration of congression as a proxy for the likelihood of initiation for each sister kinetochore pair relative to their initial distance from the nearest centrosome. This analysis revealed that the likelihood of congression initiation increases with kinetochore distance from the centrosome, particularly in the absence of CENP-E and to a lesser extent after its acute reactivation (Supplementary Fig. 1c, d). These findings suggest that centrosomes inhibit the initiation of congression when CENP-E is absent.

To better understand how centrosomes inhibit chromosome congression, we used a large-scale imaging assay to systematically study the interplay between key molecules at kinetochores and centrosomes. Using the advanced lattice light-sheet microscopy approach we recently described[11] (Methods), we captured high-resolution time-lapse images of live mitotic RPE-1 cells with labeled centromeres and centrosomes. We combined this imaging approach with CENP-E perturbation techniques that allowed us to vary CENP-E activity: CENP-E depletion, CENP-E inhibition, and control DMSO-treated cells[11] (Fig. 1a). If centrosomes inhibit the congression of polar chromosomes in the absence of CENP-E, as we hypothesized, then depleting centrioles would allow these chromosomes to initiate congression independently of CENP-E. To deplete centrioles, we inhibited polo-like kinase 4 (Plk4) activity using centrinone, a small molecule inhibitor that blocks the centriole duplication cycle[19]. By varying the duration of treatment, we generated spindles with different centriole numbers at each pole, 2:2 without treatment, 1:1 after one day, 1:0 after two days, and 0:0 after more than three days, and combined these conditions with CENP-E perturbations (Fig. 1a, b).

As a proxy for congression efficiency, we quantified the number of polar chromosomes 5 minutes after spindle elongation, the residence time of polar chromosomes at the spindle poles, and the total duration of mitosis (Methods). Following CENP-E perturbations, spindles with a 1:1 or 2:2 centriole configuration had an average of about eight polar chromosomes per spindle, compared to fewer than one chromosome under control conditions in both spindle types (Fig. 1c, d, Supplementary Video 1). In 1:0 spindles after CENP-E perturbations, the centriolar poles (labeled "1") contained approximately four polar chromosomes, similar to their counterparts in 1:1 spindles (Fig. 1c, d, Supplementary Video 1). The acentriolar poles (labeled "0") in 1:0 spindles consistently had an average of around one polar chromosome (Fig. 1c, d, and Supplementary Video 1). In 1:0 spindles, the initial appearance of polar chromosomes at both the centriolar (labeled "1") and acentriolar (labeled "0") poles was independent of CENP-E activity (Fig. 1d). However, their persistence and efficient congression depended critically on CENP-E specifically at centriolar poles. This is evidenced by the significantly longer retention of polar chromosomes at centriolar poles compared to acentriolar poles across all spindle types following CENP-E inhibition or depletion (Fig. 1e), resulting in a marked prolongation of mitosis (Supplementary Fig. 1e). Furthermore, completely acentriolar 0:0 spindles, which were rarely observed due to p53-dependent arrest in RPE-1 cells[19], exhibited a low number of polar chromosomes under CENP-E inhibition, similar to the acentriolar poles in 1:0 spindles (Fig. 1c, f, Supplementary Video 1). These findings suggest that centrioles specifically limit the initiation of polar chromosome congression when CENP-E activity is compromised.

The low number of polar chromosomes at acentriolar poles may reflect the transient monopolarization of 1:0 spindles during early mitosis[26]. However, forced transient monopolarization via Eg5 inhibitor washout, followed by CENP-E inhibition, did not change the number of polar chromosomes compared to untreated 2:2 spindles (Supplementary Fig. 2a)[11]. This suggests that the inhibitory effect of centrioles on congression initiation without CENP-E is independent of passage through a transient monopolar state. To examine microtubule organization at centriolar versus acentriolar poles, we used stimulated emission depletion (STED) super-resolution microscopy, but observed no substantial differences aside from the absence of astral microtubules at acentriolar poles (Fig. 1g). Taken together, these findings indicate that in non-transformed cells, a chromosome's ability to initiate congression when CENP-E is inactive decreases inversely with its proximity to spindle poles containing centrioles.

### Centrioles regulate congression initiation without affecting congression movement dynamics

To determine whether the inhibitory effect of centrioles on congression depends on chronic Plk4 inhibition, we examined whether centriole dislocation during prolonged mitosis could trigger congression of polar chromosomes independently of long-term Plk4 effects. We focused on rare instances (<10%, $n = 5/55$ cells) in RPE-1 cells where a centriole spontaneously and acutely dislocated from the 1:1 spindle pole during CENP-E depletion (Supplementary Fig. 2b). This phenomenon has been previously observed in prolonged mitosis[27,28]. Fascinatingly, in all cases, polar chromosomes initiated congression almost immediately following centriole displacement in CENP-E–inhibited cells (Supplementary Fig. 2b). These observations suggest that the mere presence of centrioles limits congression of polar chromosomes in the absence of CENP-E.

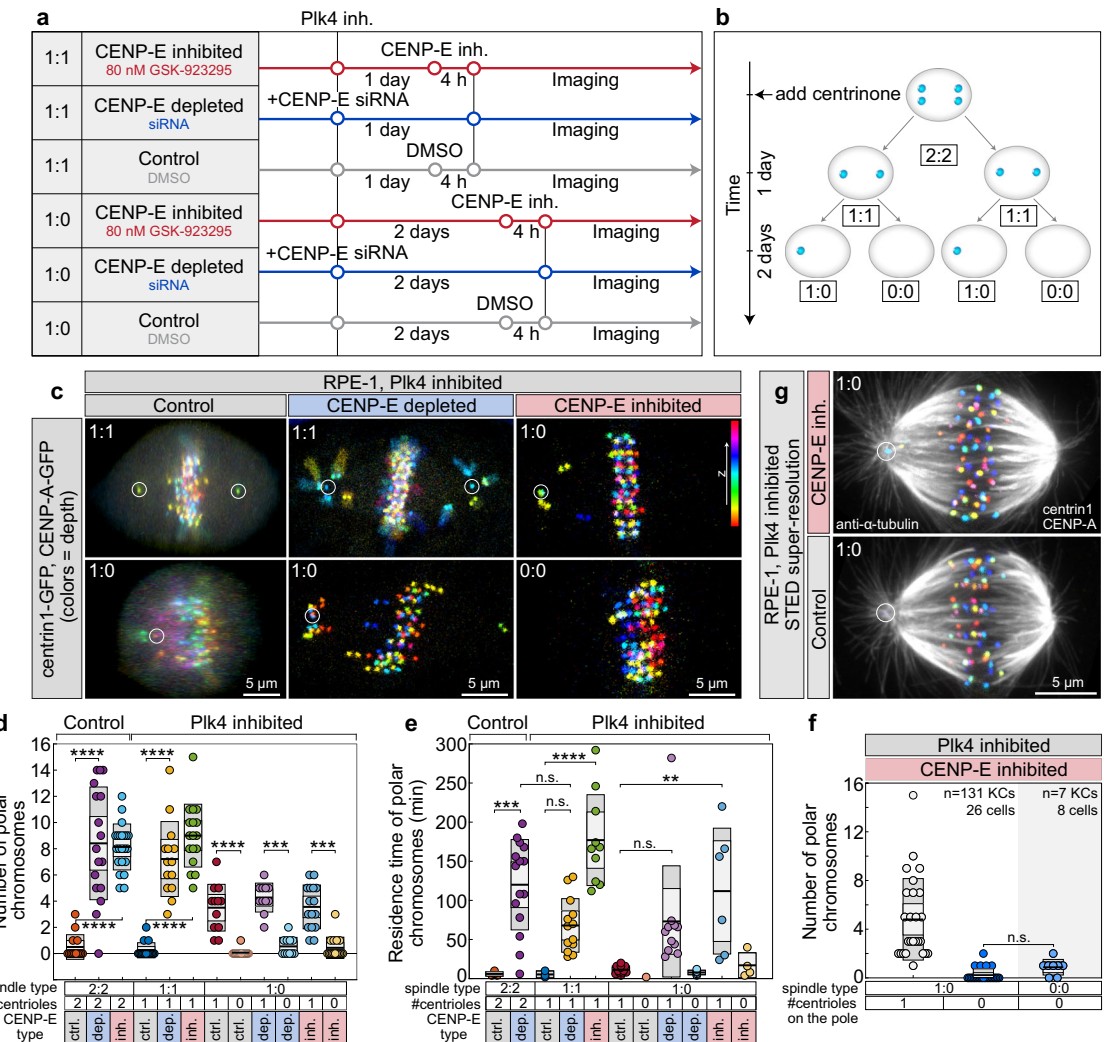

**Fig. 1 | Centrioles inhibit congression initiation when CENP-E is inactive.**
**a** Schematic of the protocol used to modulate CENP-E activity in cells with variable centriole numbers per spindle pole, generated by continuous Plk4 inhibition. Numbers on the left denote number of centrioles per spindle pole. **b** Diagram showing progressive centriole depletion after continuous Plk4 inhibition by 300 nM centrinone. **c** Representative images of live RPE-1 cells expressing CENP-A-GFP and centrin1-GFP (color-coded by depth, color bar) with varying centriole counts (white circles), 5 minutes after end of prometaphase spindle elongation, under indicated treatments. **d**, **e** Quantification of polar chromosome number **d**, and their residence time at spindle poles **e**, after prometaphase spindle elongation, comparing cells with different centriole numbers. Category "1" includes centriolar poles from both 1:1 and 1:0 spindles. Colored points represent individual cells; black lines show the mean, with light and dark gray areas marking 95% confidence intervals for the mean and standard deviation, respectively. Number of cells per condition: 14, 15, 22, 16, 14, 18, 16, 16, 12, 12, 16, 16 (**d**, **e**), each pooled from ≥3 independent biological replicates. **f** Number of polar chromosomes after treatments in cells with varying centriole numbers, fixed prior to measurement. Pooled data from ≥3 independent biological replicates. Number of kinetochores and cells is given in the figure. Dispersion measures as in (**d**, **e**). **g** STED microscopy images of RPE-1 cells immunostained for α-tubulin (gray), expressing CENP-A-GFP and centrin1-GFP (color-coded by depth). All images are maximum projections. Statistics: two-tailed ANOVA with post-hoc Tukey's HSD test. Symbols indicate: n.s., $p > 0.05$; *, $p \leq 0.05$; **, $p \leq 0.01$; ***, $p \leq 0.001$; ****, $p \leq 0.0001$; inh., inhibited; depl., depleted; siRNA, small interfering RNA; KC, kinetochore; ctrl., control. Source data are provided as a Source Data file.

Having found that centrioles inhibit congression initiation in the absence of CENP-E, we next asked whether centriole number influences the dynamics of congression movement. Congression velocity and the increase in interkinetochore distance were indistinguishable among 1:1, 1:0, and 2:2[11] spindles following CENP-E inhibition in RPE-1 cells (Fig. 2a, Supplementary Fig. 2c–g). These results suggest that centrioles primarily inhibit the initiation of congression but do not significantly influence the subsequent movement of polar chromosomes toward the metaphase plate.

To test whether selective disruption of CENP-E activity, without inducing complete depletion or rigor microtubule binding of the motor, would bias the accumulation of polar chromosomes to the centriolar pole, we used osteosarcoma U2OS cells. These cells were engineered for doxycycline-inducible expression of a phospho-null

Threonine 422 (T422A) mutant, which lacks the Aurora A/B-specific phosphorylation site[23]. To compare different modes of CENP-E perturbation in transformed cells, we analyzed three conditions: 1) endogenous CENP-E depletion, 2) pharmacological inhibition of CENP-E, and 3) expression of the T422A mutant following depletion of endogenous CENP-E. Immunofluorescence microscopy was performed after continuous centrinone treatment for 2 days to generate mixed populations of cells with either centriolar (1:1) or mixed centriolar and acentriolar (1:0) spindle poles (Fig. 2b). We quantified the number of polar chromosomes relative to the number of centrioles on the spindle pole. Consistent with our findings in RPE-1 cells (Fig. 1d, e), polar chromosomes predominantly accumulated at centriolar poles under all conditions that impaired CENP-E activity, including expression of the T422A mutant (Fig. 2c). These results demonstrate that

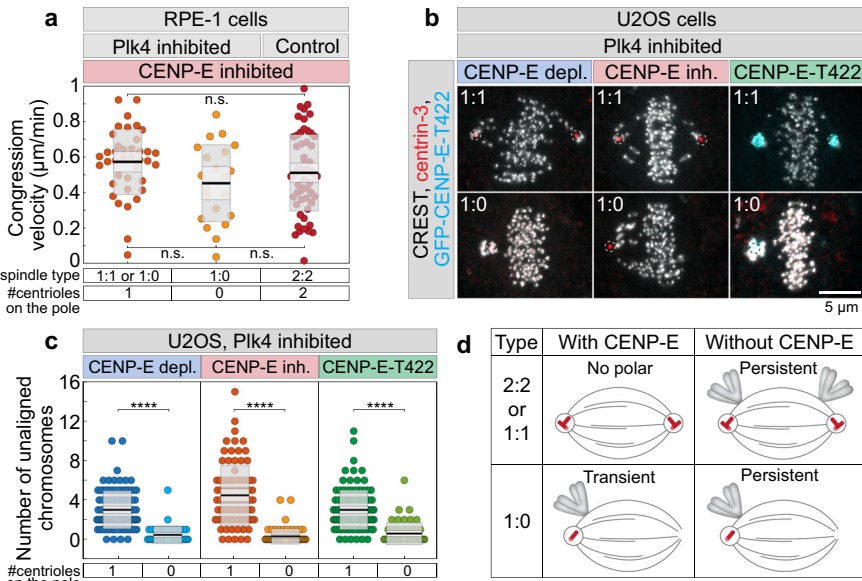

**Fig. 2 | Centrioles inhibit the initiation of congression in cells expressing phospho-null CENP-E, without affecting congression velocity. a** Velocity of chromosome congression in RPE-1 cells during the 6-minute period preceding complete alignment for each initially polar kinetochore pair, across different treatments and centriole numbers. Colored points represent individual cells; black lines show the mean, with light and dark gray areas marking 95% confidence intervals for the mean and standard deviation, respectively. **b** Representative examples of fixed U2OS cells induced to express GFP-CENP-E-T422 (cyan) and immunostained for human centromere protein (CREST, gray) and centrin-3 (red, dashed circles), with different numbers of centrioles on each pole (rows) and for different treatments (columns) as indicated on the top. **c** Number of polar chromosomes in U2OS cells after different treatments, as indicated on top, and with different numbers of centrioles, as indicated on bottom. Dispersion measures as in (**a**). **d** Schematic summarizing experimental results for the appearance and residence time (transient or persistent) of polar chromosomes at spindle poles in cells with varying centriole numbers, with or without CENP-E activity. All images are maximum projections. Numbers: **a** 39, 21 and 60 kinetochore pairs from 36 cells, **c** 100, 44, 84, 20, 104, 51 cells, each pooled from ≥2 independent biological replicates. Category "1" includes centriolar poles from both 1:1 and 1:0 spindles. Statistics: two-tailed ANOVA with post-hoc Tukey's HSD test. Symbols indicate: n.s., non-significant; ****, $p \leq 0.0001$; inh., inhibited; depl., depleted. Source data are provided as a Source Data file.

CENP-E-dependent chromosome congression is biased toward centriolar spindle poles, both in non-transformed and transformed cells and across distinct modes of CENP-E perturbation (Fig. 2d).

## Hyperactivated Aurora B near centrosomes increases KMN phosphorylation and delays congression without CENP-E

Which molecular factors delay congression initiation near centrioles in the absence of CENP-E? Since we have shown that Aurora A inhibition promotes congression of polar chromosomes without CENP[11], we hypothesized that centrosomal Aurora A activity prevents chromosome congression at centriolar poles when CENP-E is absent. To test whether Aurora A activity differs between centriolar and acentriolar spindle poles, we quantified active Aurora A (pAurA, pT288-Aurora A) at spindle poles with varying centriole numbers, generated by continuous Plk4 inhibition in RPE-1 cells, under both control and CENP-E-inhibited conditions. Intriguingly, we found that acentriolar poles exhibited ~4-fold lower pAurA levels than centriolar poles, whereas the maximum difference between poles in 1:1 spindles was only ~10% (Fig. 3a, b). These differences were independent of CENP-E (Fig. 3a, b), indicating that CENP-E does not regulate Aurora A activity. Together with our observation that Aurora A inhibition promotes congression in the absence of CENP-E[11], these results suggest that polar chromosomes depend on CENP-E for congression due to high Aurora A activity near centrosomes.

We hypothesized that high Aurora A activity at centriolar spindle poles increases activation of kinetochore-localized Aurora B on polar chromosomes. Elevated Aurora B would increase phosborylation of its downstream targets, such as Knl1, a key KMN network component[29], thereby blocking congression. Indeed, we have observed that the mean signals of pAurB (pT232-Aurora B) and pKnl1 (pS24, pS60-Knl1)[30] on polar kinetochores near centriolar poles were significantly higher than on aligned kinetochores or the rare polar kinetochores at acentriolar poles under CENP-E inhibition (Fig. 3c–f, Supplementary Fig. 3a, b).

To confirm the specificity of phospho-antibodies as indicators of Aurora B activity, we acutely inhibited Aurora B with 3 µM ZM-447439 and assessed pAurB and pKnl1 localization and intensity. Both signals were nearly abolished at kinetochores, except in a small centriolar region, confirming their specificity as Aurora B activity markers (Supplementary Fig. 3c). In contrast, the intensity and localization of pHec1 (pS55-Hec1)[31,32] remained unchanged following acute Aurora B inhibition (Supplementary Fig. 3c), suggesting that S55 phosphorylation does not specifically reflect Aurora B activity, consistent with a recent report[33]. This likely explains the uniform pHec1 levels at polar kinetochores regardless of centrioles on spindle poles or CENP-E activity (Supplementary Fig. 3d, e), in contrast to the spatial variation and CENP-E dependence observed for pKnl1 and pAurB (Fig. 3c–f). In DMSO-treated control cells, pAurB and pKnl1 levels were similar between aligned and rare polar kinetochores, and significantly lower than in the same kinetochore groups under CENP-E inhibition (Fig. 3c–f).

Collectively, these results suggest that in the absence of CENP-E, Aurora A at centriolar poles overactivates kinetochore-localized Aurora B. This increases phosphorylation of outer kinetochore targets such as Knl1, and likely the entire KMN network (Fig. 3g). Hyperphosphorylation of the KMN network prevents stabilization of end-on microtubule attachments at kinetochores, thereby blocking chromosome congression.

## Aurora A overactivates Aurora B on kinetochores near centrosomes

Given that polar kinetochores near centrosomes in the absence of CENP-E show elevated Aurora B phosphorylation, we hypothesized

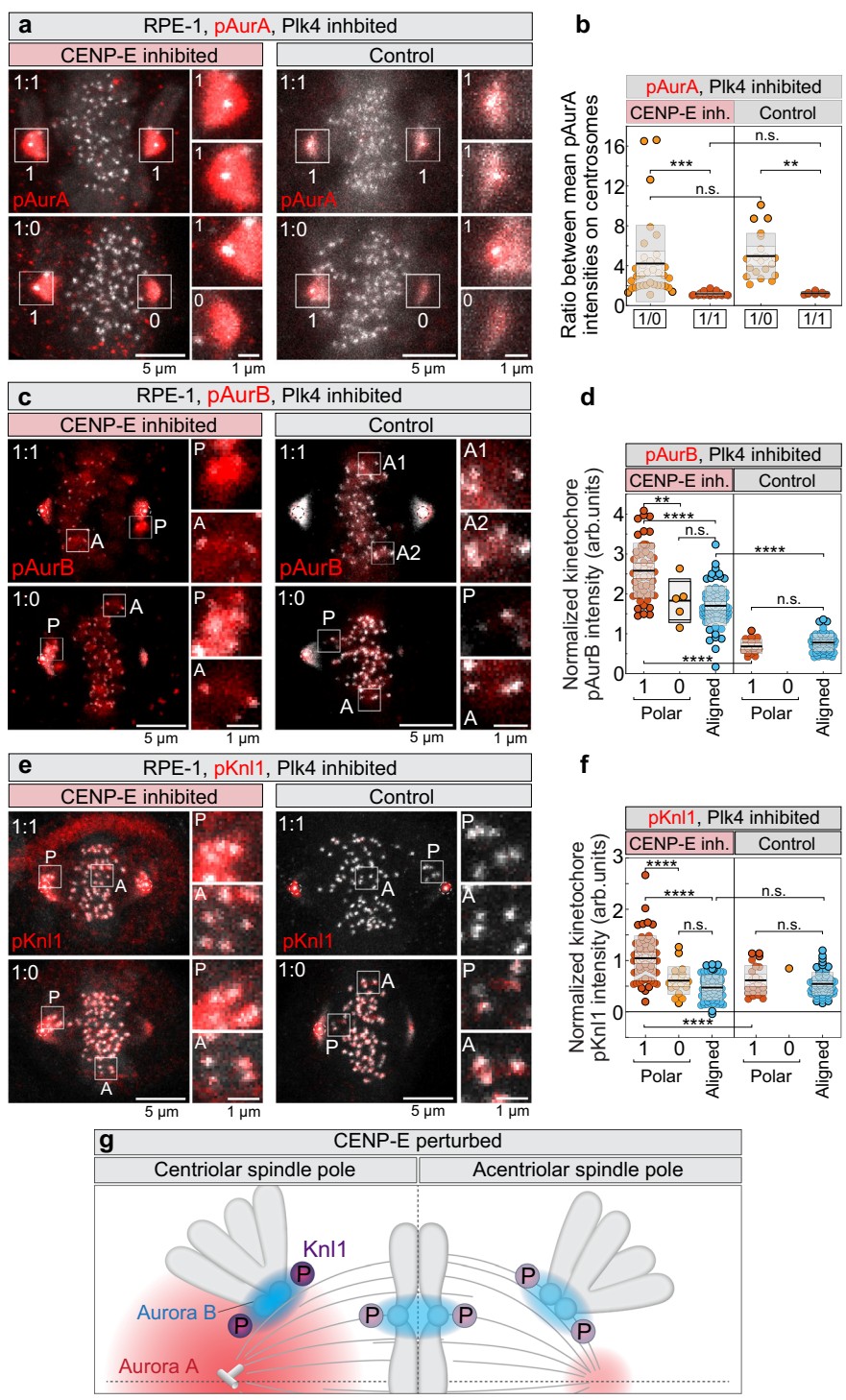

that Aurora A at spindle poles directly promotes Aurora B activity at kinetochores, consistent with their shared substrate recognition motifs[34–36]. To test this, we first inhibited CENP-E and then acutely inhibited Aurora A by the highly specific inhibitor MLN8054[37], using a range of concentrations and treatment durations. We then compared pAurB levels at kinetochores in CENP-E-inhibited cells following acute treatment with Aurora A inhibitor or DMSO (Fig. 4a). Acute inhibition of Aurora A significantly reduced Aurora B activity at polar kinetochores, lowering it to the levels seen on aligned kinetochores in CENP-E–inhibited cells (Fig. 4a–c). To validate this finding, we tested two additional Aurora A–specific inhibitors: TCS7010[38], which also reduced pAurB asymmetry on polar versus aligned kinetochores similar to MLN8054, and Alisertib[39], which had

no significant impact under the tested conditions (Fig. 4d, e). Consistent with these results, only MLN8054 and TCS7010 caused a ~ 2-fold decrease in the number of polar chromosomes and a significant reduction in spindle length[40] compared to CENP-E inhibition alone, without altering pAurB levels at aligned kinetochores (Fig. 4f, g, Supplementary Fig. 3f). These findings support a model in which Aurora A increases Aurora B activity at polar kinetochores when CENP-E is inactive.

Our model suggests that downregulation of Aurora kinases promotes chromosome congression by stabilizing end-on attachments at polar kinetochores. To test this, we stained tubulin in cells pre-treated with a CENP-E inhibitor, followed by acute treatment with the Aurora A inhibitor MLN8054 (Fig. 4h).

**Fig. 3 | High Aurora A activity at centriolar poles drives Aurora B and KMN network hyperphosphorylation when CENP-E is inactive. a, c, e** Representative RPE-1 cells expressing CENP-A-GFP and centrin1-GFP (gray) with centrioles marked by white circles, immunostained for phosphorylated residues of Aurora A (pT288, **a**), Aurora B (pT232, **c**), and Knl1 (pS24, pS60, **e**) (red) after the indicated treatments in cells with distinct centriole numbers. Images are maximum projections with enlarged insets of polar (P) and aligned (A) kinetochores on the right. **b, d, f** Quantification of the ratio of phosphorylated Aurora A intensity at centriolar versus acentriolar spindle poles **b**, and normalized phosphorylation levels of Aurora B **d** and Knl1 **f** at the kinetochores under different treatments in cells with distinct centriole numbers. Colored points represent individual cells; black lines show the mean, with light and dark gray areas marking 95% confidence intervals for the mean and standard deviation, respectively. **g** Schematic representation of phosphorylation levels of kinetochore components in the absence of CENP-E

activity. Aurora B activity is depicted as a blue gradient centered on the centromere, with intensity scaling according to phosphorylation level. Knl1 phosphorylation (P, circled) is represented by the intensity of purple circles. The spindle features one centriolar pole (left; centrioles shown schematically) and one acentriolar pole (right). The Aurora A activity gradient (red) is stronger and extends further from the centriolar pole. Phosphorylation patterns are illustrated for both polar and aligned chromosomes. Dashed lines indicate spindle main axes. Numbers: **b** 35, 23, 20, 10 cells, **d** 52, 5, 77, 18, 60 kinetochore pairs from 42 cells, and **f** 59, 20, 61, 22, 1, 64 kinetochore pairs from 63 cells, each pooled from ≥2 independent biological replicates. Category "1" includes centriolar poles from both 1:1 and 1:0 spindles. Statistics: two-tailed ANOVA with post-hoc Tukey's HSD test. Symbols indicate: n.s., $p > 0.05$; **, $p \leq 0.01$; ***, $p \leq 0.001$; ****, $p \leq 0.0001$; inh., inhibited; arb., arbitrary. Source data are provided as a Source Data file.

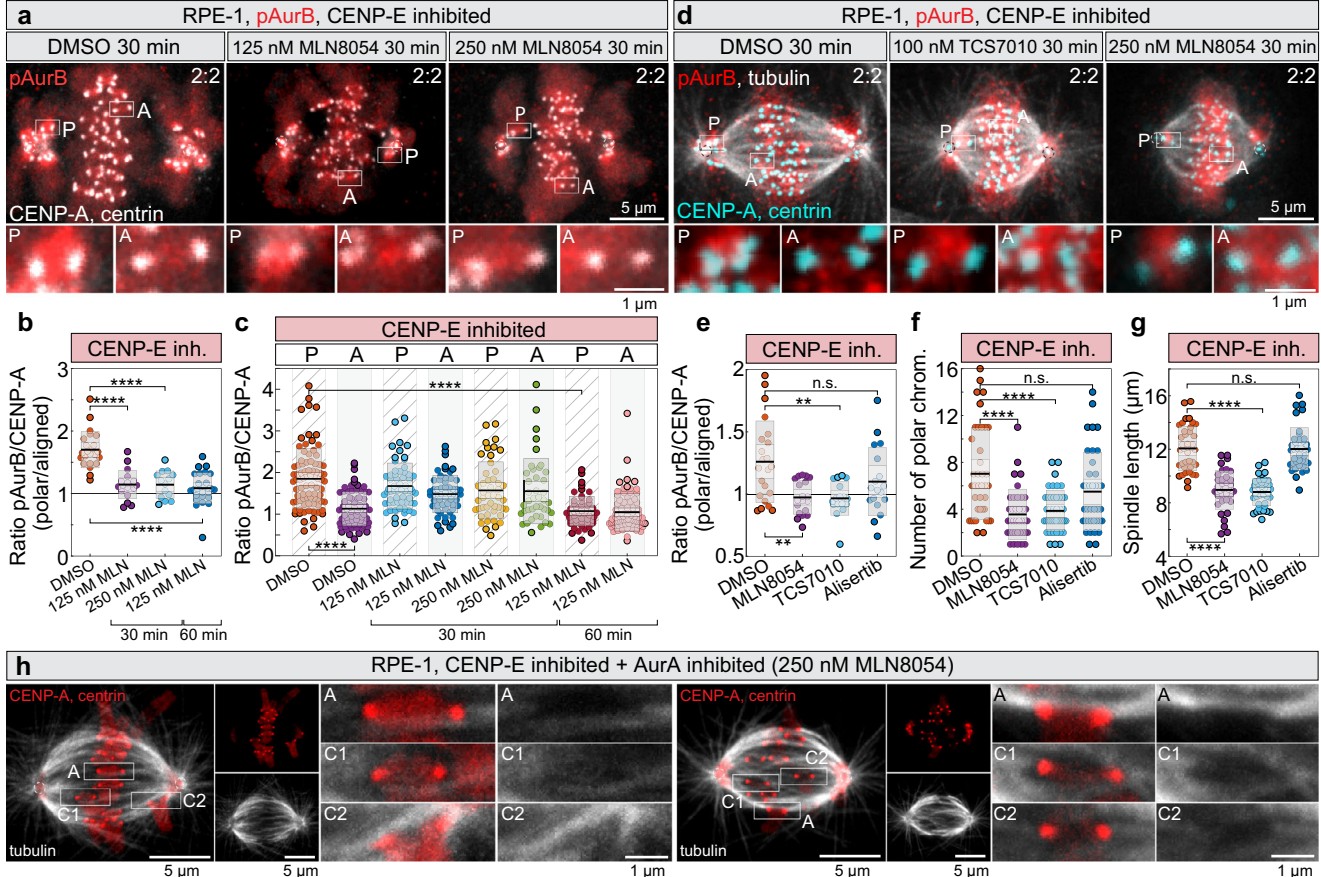

**Fig. 4 | Aurora A on centrosomes upregulates Aurora B activity on kinetochores. a** Representative RPE-1 cells expressing CENP-A-GFP and centrin1-GFP (gray, centrioles circled) after indicated treatments, showing enlarged insets of polar (P) and aligned (A) kinetochores immunostained for pT232-Aurora B (red). **b** Ratio of pT232-Aurora B intensity normalized to CENP-A on polar vs. aligned kinetochores per cell under indicated treatments. Colored points represent individual cells; black lines show the mean, with light and dark gray areas marking 95% confidence intervals for the mean and standard deviation, respectively. **c** Average pT232-Aurora B levels normalized to CENP-A on all kinetochore groups after indicated treatments. Dispersion measures as in (**b**). **d** Representative cells expressing CENP-A-GFP and centrin1-GFP (cyan, centrioles circled) immunostained for pT232-Aurora B (red) and α-tubulin (gray) after continuous CENP-E inhibition and acute Aurora A inhibition by MLN8054 and TCS7010, with enlarged insets of polar (P) and

aligned (A) kinetochores. **e–g** Quantification of pT232-Aurora B intensity ratio on polar vs. aligned kinetochores **e**, number of polar chromosomes **f**, and spindle length **g** after indicated treatments. Dispersion measures as in (**b**). **h** RPE-1 cells expressing CENP-A-GFP and centrin-GFP (red, centrioles circled), immunostained for α-tubulin (gray), and imaged by super-resolution Airyscan microscopy showing enlarged insets of congressing (C) and aligned (A) kinetochores. Images are deconvolved projections of 2–5 z-planes. Numbers: **b** 23, 18, 14, 27 cells averaged from 491 kinetochore pairs, **c** 83, 71, 61, 57, 47, 41, 52, 79 kinetochore pairs from 91 cells, **e** 22, 19, 17, 18 cells averaged from 436 kinetochore pairs, **f, g** 39, 38, 42, 38 cells, all pooled from ≥2 independent biological replicates. Statistics: two-tailed ANOVA with post-hoc Tukey's HSD test. Symbols indicate: n.s., $p > 0.05$; **, $p \leq 0.01$; ***, $p \leq 0.001$; ****, $p \leq 0.0001$; inh., inhibited; chrom., chromosomes; MLN, MLN8237. Source data are provided as a Source Data file.

Super-resolution Airyscan imaging[41] revealed that most kinetochores located between the centrosome and the metaphase plate had end-on microtubule attachments after acute MLN8054 treatment in CENP-E–inhibited cells ($n = 15$ cells) (Fig. 4h,

Supplementary Fig. 3g). These results are consistent with our finding that acute CENP-E reactivation stabilizes end-on attachments on congressing chromosomes[11]. Collectively, these findings indicate that Aurora A modulates Aurora B activity near spindle poles,

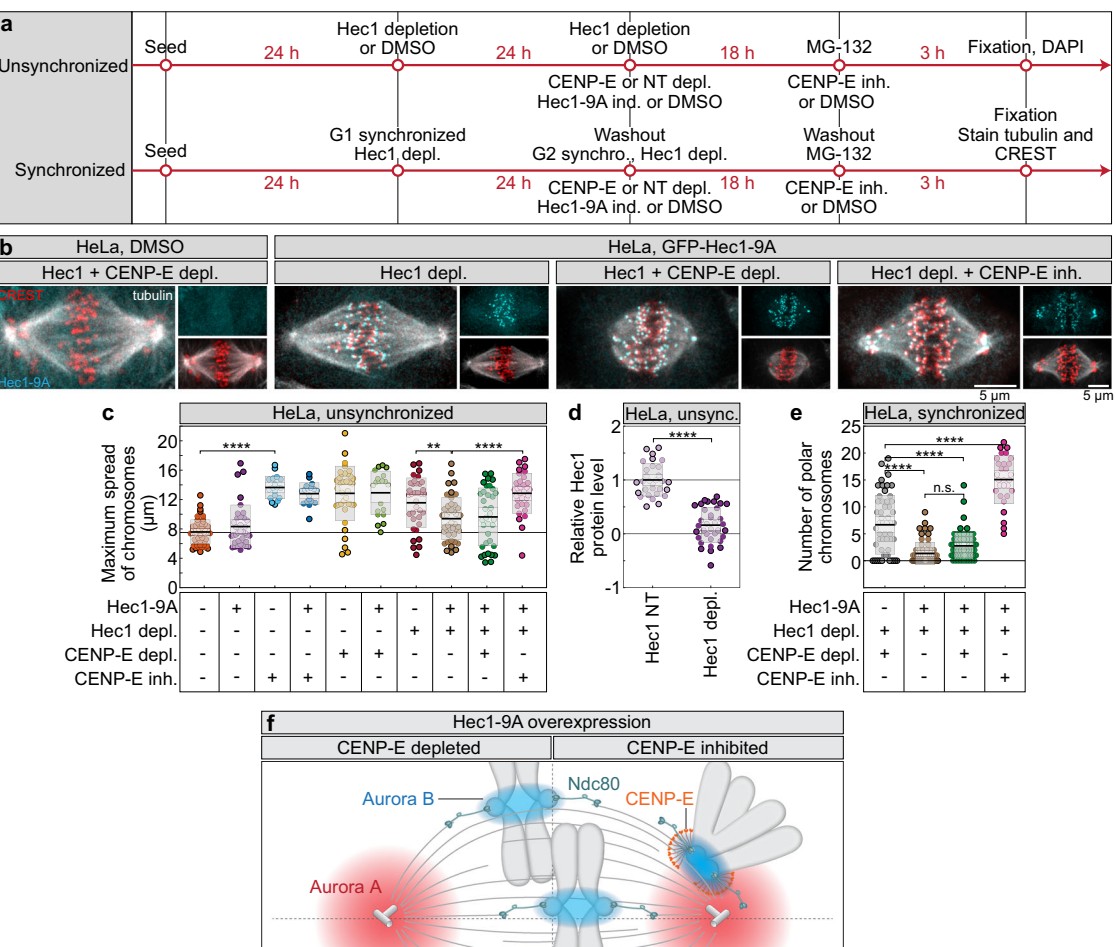

**Fig. 5 | Aurora B kinase–mediated Hec1 phosphorylation impairs chromosome congression in the absence of CENP-E. a** Schematics of two experimental work-flows using either unsynchronized HeLa cells (top) or cells synchronized in G1 and G2 during protein depletions (bottom). Treatments above the line are common to all conditions, while those below the line are specific to each subgroup.
**b** Representative images of HeLa cells synchronized in mitosis with MG-132, with-out (left) or with (all others) GFP-Hec1-9A expression (cyan), immunostained for α-tubulin (gray) and centromeres (CREST, red) in treatments indicated on top.
**c** Maximum chromosome spread in unsynchronized cells after indicated treat-ments. Colored points represent individual cells; black lines show the mean, with light and dark gray areas marking 95% confidence intervals for the mean and standard deviation, respectively. In **b** and **c** MLN stands for MLN8054. **d** Hec1 levels measured within the DAPI-stained DNA region after non-targeting (NT) or Hec1 3′UTR depletion, normalized to NT control group average. Dispersion measures as in (**c**). **e** Number of polar chromosomes in synchronized HeLa cells after indicated treatments. Dispersion measures as in (**c**). **f** Schematic of chromosome congression

in cells overexpressing Hec1-9A (represented as green element, Ndc80) with endogenous Hec1 depleted. In CENP-E–depleted cells (left), congression is efficient, with Hec1-9A forming stable end-on microtubule attachments, similar to aligned kinetochores (bottom). In contrast, under CENP-E inhibition (right), the fibrous corona (simplified as inactive CENP-E, represented as orange motor) blocks Hec1–microtubule attachments, impairing congression at polar kinetochores. Aurora B activity is shown as a blue gradient centered on the centromere, while Aurora A is represented as a red gradient centered on a pair of schematically depicted centrioles. Dashed lines indicate spindle main axes. Numbers: **c** 53, 36, 18, 18, 36, 20, 35, 44, 33, 32, 35 cells, **d** 30, 35 cells, and **e** 53, 62, 70, 29 cells, all pooled from ≥3 independent biological replicates. Statistics: two-tailed ANOVA with post-hoc Tukey's HSD test. Symbols indicate: n.s., $p > 0.05$; **, $p \leq 0.01$; ****, $p \leq 0.0001$; inh., inhibited; depl., depleted; NT, non-targeting; ind., induced; synchro., synchronization; unsync., unsynchronized. Source data are provided as a Source Data file.

supporting our model of localized regulation of chromosome congression.

If centrosomal Aurora A enhances Aurora B activity in the absence of CENP-E, then Aurora B–mediated phosphorylation of outer kinetochore proteins should decrease as chromosomes move away from the centrosome during congression. Indeed, we observed that the level of pKnl1 was inversely correlated with the distance of the kinetochore pair from the centrosome in the absence of CENP-E activity (Supplementary Fig. 3h). In conclusion, chromosome congression is accompanied by a reduction in Aurora B activity on kinetochores, leading to decreased phosphorylation of KMN network components. The reduction of Aurora B activity on congressing chromosomes is crucial for the stable attachment of microtubules to kinetochores[42] and is significantly accelerated by the motor activity of CENP-E.

## Constitutive Hec1 dephosphorylation is sufficient to induce congression without CENP-E
We have shown that in the absence of CENP-E, Aurora A hyperactivates Aurora B on polar chromosomes, leading to hyperphosphorylation of downstream targets such as the essential microtubule end-binding protein Hec1. Based on this, we hypothesized that blocking Hec1 phosphorylation would enable congression initiation without CENP-E. To test if downregulation of Aurora kinase-mediated Hec1 phosphorylation at polar kinetochores is sufficient to initiate chromosome congression, we used HeLa cells engineered to express an inducible, siRNA-resistant GFP-Hec1-9A phospho-mutant[43] (Fig. 5a, top, b). This mutant mimics the constitutive dephosphorylation of Hec1 at nine phosphorylation sites, which are phosphorylated by Aurora B kinase[44]. Expression of Hec1-9A in the presence of endogenous Hec1 mildly affected chromosome congression but increased interkinetochore

distance of aligned kinetochore pairs (Fig. 5b, c, Supplementary Fig. 3i), consistent with previous reports[43,44]. Chromosome congression was completely disrupted by depletion or inhibition of CENP-E, as well as Hec1 depletion, leading to an accumulation of polar chromosomes, consistent with previous studies (Fig. 5b-d)[8,45]. Interestingly, both depletion and inhibition of CENP-E caused severe congression defects in cells expressing endogenous Hec1, even upon Hec1-9A overexpression (Fig. 5c). This suggests that Hec1-9A overexpression cannot compensate for CENP-E disruption in the presence of endogenous Hec1, despite its dominant-negative effect on interkinetochore tension and SAC satisfaction[43,46].

Based on our finding that acute inhibition of Aurora kinases induces chromosome congression independently of CENP-E[11], we hypothesized that constitutive Hec1 dephosphorylation would compensate for the loss of CENP-E following endogenous Hec1 depletion. To control for potential mitotic prolongation effects from combined Hec1 and CENP-E perturbations, we synchronized cells in G1 to ensure depletion of endogenous Hec1, CENP-E, or both before mitotic entry. After release into G2 and mitosis, cells were arrested in metaphase using a proteasome inhibitor for a maximum of 2 h (Fig. 5a, bottom). We compared this assay with experiments in which depletion was carried out in unsynchronized cells (Fig. 5a, top).

We assessed chromosome congression efficiency by immunostaining cells for tubulin and centromeres under four conditions, all in the context of endogenous Hec1 depletion: (1) CENP-E depletion, (2) Hec1-9A expression, (3) Hec1-9A expression with CENP-E depletion, and (4) Hec1-9A expression with CENP-E inhibition. Expression of Hec1-9A in cells depleted of endogenous Hec1 rescued major chromosome congression defects in both synchronized and unsynchronized cells (Fig. 5b, c, e), as previously reported[43,44]. Remarkably, expression of Hec1-9A also significantly improved chromosome congression in cells depleted of both endogenous Hec1 and CENP-E (Fig. 5b). Under this condition, maximum chromosome spread and average polar chromosome numbers were indistinguishable from those in CENP-E–intact cells, in both synchronized (Fig. 5b, e; Supplementary Fig. 3j) and unsynchronized cells (Fig. 5c; Supplementary Fig. 3i). In all conditions with Hec1-9A overexpression, aligned chromosomes exhibited increased interkinetochore distances and formed end-on microtubule attachments (Fig. 5b). Surprisingly, Hec1-9A overexpression did not rescue the congression defects caused by CENP-E inhibition in either synchronized or unsynchronized cells (Fig. 5b, c, e). This is likely due to hyperexpansion of the fibrous corona on polar kinetochores following CENP-E inhibition, which interferes with the stabilization of microtubule attachments[11]. Consistent with this, polar kinetochores in Hec1-9A-overexpressing cells under CENP-E inhibition (Fig. 5b) were positioned near spindle poles with low interkinetochore distance, resembling those under CENP-E inhibition alone (Fig. 4d), suggesting a lack of end-on attachments. Altogether, these findings indicate that constitutive Hec1 dephosphorylation is sufficient to drive chromosome congression without CENP-E and endogenous Hec1, but only when fibrous corona expansion is limited.

## Discussion

In this study, we propose that the timing of chromosome movement is regulated by the interplay between CENP-E motor activity and Aurora B kinase activity at kinetochores, together with chromosome proximity to centrosomes, which determines their exposure to the Aurora A gradient (Fig. 6). Previous research has shown that Aurora kinases phosphorylate and activate CENP-E near centrosomes[10,23]. Our recent results[11], together with the results shown here, demonstrate that Aurora kinases inhibit the initiation of chromosome congression in the absence of CENP-E. To reconcile these seemingly conflicting roles of Aurora kinases within the same process, we propose a feedback loop where activity of kinetochore components is influenced by the proximity to centrosomes, where the Aurora A gradient is concentrated in

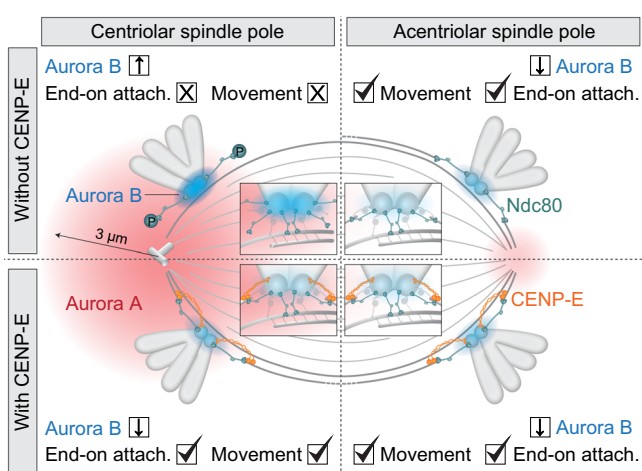

**Fig. 6 | Spatial regulation of chromosome congression via Aurora A–mediated control of kinetochore activity.** The speed of end-on kinetochore–microtubule attachment formation and chromosome congression depends on chromosome–centrosome distance. Without CENP-E (orange motor), polar chromosomes near centriolar centrosomes (top left) remain laterally attached to microtubules due to high Aurora A (AurA, red gradient, ~3 μm radius) activity, which enhances Aurora B (AurB, blue gradient) activation. This causes hyperphosphorylation of outer kinetochore proteins (simplified as Ndc80, green) and expansion of the fibrous corona (not shown), preventing stabilization of end-on attachments. With CENP-E (bottom left), stabilization of initial end-on attachments leads to reduced AurB activity, allowing complete end-on conversion, biorientation, and congression. Without centrosomes and CENP-E (top right), lower AurA leads to reduced AurB activity, enabling end-on conversion and congression independently of CENP-E. With CENP-E at kinetochores near acentriolar poles (bottom right), congression resembles that at centriolar poles. Insets illustrate kinetochore-level details. Symbols indicate: X = absence or low activity; ↓ = downregulation; ↑ = upregulation; √ = presence; attach. = attachment.

somatic human cells (Fig. 6)[47]. Without CENP-E, Aurora A overactivates Aurora B and its targets, such as KMN complex[13,30,42,44] and the fibrous corona components[48,49], inhibiting end-on attachment stabilization and congression[8,11,50] (Fig. 6). However, direct biochemical evidence for an Aurora A–Aurora B interaction has yet to be established. Aurora A also directly phosphorylates KMN components, including Hec1, near centrioles[32,50]. The specific contributions of Aurora kinases to Hec1 phosphorylation sites (S44, S55, S69, and others)[32,33] and the role of distinct phosphorylation sites in congression inhibition remain unclear. Future studies will hopefully clarify this.

Our findings support a model in which kinetochore-centrosome feedback is not solely driven by the attachment status of microtubules to kinetochores but involves a bidirectional regulatory loop between Aurora kinases and CENP-E. While CENP-E promotes stable end-on attachments that suppress Aurora B activity[11], Aurora A and B themselves increase CENP-E activity and localization[10,23]. This interdependence suggests that changes in kinetochore signaling represent not merely passive responses to attachment status but active feedback between motor activity and kinase signaling near the spindle poles. When CENP-E is present at kinetochores, it promotes the stabilization of end-on attachments, which reduces Aurora B activity and consequently facilitates fibrous corona removal and effective congression initiation (Fig. 6)[11]. If the Aurora A gradient is weakened, as in acentriolar spindle poles or after acute Aurora A inhibition[11], Aurora B activation decreases, allowing end-on attachment and CENP-E-independent congression of polar chromosomes (Fig. 6). However, weak activity of both Aurora kinases is linked to limited correction of aberrant kinetochore–microtubule attachments, which can result in excessive chromosome mis-segregation during anaphase[19,50,51].

During unperturbed mitosis in healthy human cells, chromosomes typically remain at least 3 μm from the centrosome and achieve rapid biorientation outside this region[16,25,52]. Our model suggests that when CENP-E is active, both polar and non-polar chromosomes rapidly establish end-on attachments and become bioriented near the spindle surface by capturing microtubules from the opposite spindle half[16,53]. Thus, while CENP-E supports microtubule end-on stabilization at all kinetochores, as indicated by reduced microtubule density in kinetochore fibers upon its perturbation[28], its loss predominantly affects polar chromosomes without causing widespread detachment of those positioned farther from centrosomes.

The model we present clarifies the signaling dynamics between centrosomes and kinetochores during the initiation of chromosome congression. It explains why pole-proximal kinetochores within the Aurora A gradient rely on CENP-E for rapid congression during early mitosis[8]. This reliance is consistent with findings that congression is initiated by constitutive Hec1 dephosphorylation or acute Aurora kinase inhibition independently of presence of CENP-E[11], as the KMN network retains maximal microtubule affinity under these conditions[43,44]. However, our recent results[11], together with those presented here, indicate that reduction of the fibrous corona is also required for efficient stabilization of early end-on attachments and successful initiation of congression. This suggests that congression initiation involves two distinct processes that are interconnected through Aurora B kinase. Recent work further indicates that kinetochores can modulate Aurora A activity via Mps1[54], underscoring feedback between centrosomes and kinetochores.

While our study highlights the central role of CENP-E and Aurora kinases in chromosome congression, other spindle-associated proteins are likely involved. HURP and CLASP proteins, for instance, stabilize and regulate kinetochore microtubules near chromosomes[55,56], and the Ska complex enhances kinetochore–microtubule coupling under tension[57]. At the spindle poles and in their vicinity, factors such as TPX2, pericentriolar proteins, kinesin-13 depolymerases, the cross-linker NuMA, and the Augmin complex are all intricately linked to Aurora A signaling[36]. These pathways may act downstream of CENP-E and Aurora kinases to influence both congression efficiency and spatial bias, meriting further investigation.

Our findings suggest that it is not the presence of centrioles per se, but rather elevated Aurora A kinase activity that governs the feedback dynamics between spindle poles and kinetochores. In somatic cells, Aurora A activity is associated with canonical centrosomes, whereas oocytes maintain Aurora A activity and form functional spindles without canonical centrosomes[58]. This may explain why oocytes, but not acentriolar poles in somatic cells, require CENP-E for chromosome congression despite the absence of centrioles[59,60]. We propose that kinetochore feedback mechanisms controlled by Aurora A may operate independently of centrioles, highlighting the need for further study of chromosome congression regulation in acentriolar contexts.

Regarding mitosis in diseased conditions, overexpression of Aurora A kinase and loss of the CHK2-BRCA1 tumor suppressor pathway, both of which lead to elevated Aurora A activity at centrosomes, are frequently observed in human cancers[61,62]. Increased Aurora A activity enhances microtubule assembly rates at kinetochores through mechanisms that are not yet fully understood, contributing to chromosomal instability in colorectal cancer cells[63]. Our findings suggest that elevated Aurora A activity could impede chromosome congression if not balanced by changes in other key components of the centrosome-kinetochore feedback loop, such as CENP-E/BubR1 and Aurora B kinase. Disruptions in this balance may explain the aberrant congression observed in colorectal cancer cells with overactive Aurora A kinase, which correlates with increased mitotic timing, and chromosome mis-segregation rates in those cells[63].

The inhibitory activity of Aurora A on chromosome congression might also explain why chromosomes near the poles experience delayed congression, not only in the absence of CENP-E but also in other contexts where CENP-E is active. For example, in multipolar spindles with supernumerary centrosomes, chromosomes near coalesced spindle poles are often delayed in congression[64], potentially due to their proximity to two centrosomes and their Aurora A gradients. Likewise, cancer cells of diverse origins frequently exhibit difficulties in congressing pole-proximal chromosomes, resulting in prolonged mitosis[65–67]. Although the underlying causes of these congression defects remain unclear, pole-proximal chromosomes are often mis-segregated into micronuclei, contributing to aneuploidy[65]. We propose that the congression defects observed across human tumors may arise from imbalances in the signaling feedback that regulates chromosome congression, thereby disrupting the stability of end-on attachments, a hallmark of chromosomally unstable human tumors[33,68].

## Limitations of the study

It remains unclear why CENP-E inhibition was not effectively rescued by overexpressing Hec1-9A in the context of endogenous Hec1 depletion, unlike with CENP-E depletion. In addition to the fact that extensive fibrous corona expansion limits Hec1–microtubule engagement upon CENP-E inhibition[49], another possibility is that the Hec1-S69 phosphorylation site, absent in the Hec1-9A mutant[44], may be essential for bypassing the need for CENP-E under these conditions. Also, as Hec1 is known as notoriously difficult protein to be fully depleted from the kinetochore, due to its role as a core, stable kinetochore component[31], some residual endogenous Hec1 might still be phosphorylated on the kinetochores. Further research is needed to clarify this. Moreover, Aurora A might have a more direct impact on KMN network components, potentially through specific phosphorylation sites on Hec1, such as S69, which were not examined in this study[32,50].

In spindles where centrioles were removed using the Plk4 inhibitor, potential indirect effects on chromosome congression cannot be fully excluded. For example, the asymmetric distribution of polar chromosomes may reflect differences in position and movement between centriolar and acentriolar poles, in addition to differences in biorientation capacity. However, a similar trend was observed, though in a smaller number of cells, following spontaneous centriole displacement from spindle poles in CENP-E-depleted cells. Moreover, live-cell imaging showed that during spindle elongation, chromosomes moved toward both poles but remained significantly longer at the centriolar pole when CENP-E was perturbed (Supplementary Video 1), supporting the robustness of our conclusions.

## Methods
### Cell lines and culture
All reagents and tools used in this study, along with their details, are provided in Supplementary Table 1. Experiments were carried out using human hTERT-RPE-1 (hTERT-immortalized retinal pigment epithelium) cells stably expressing CENP-A-GFP, human hTERT-RPE-1 cells stably expressing both CENP-A-GFP and centrin1-GFP and human hTERT-RPE-1 cells stably expressing CENP-A-GFP and Mis12-mCherry, all courtesy of Alexey Khodjakov (Wadsworth Center, New York State Department of Health, Albany, NY, USA), human HeLa (cervical carcinoma patient) cells expressing GFP-Hec1-9A, courtesy of Geert Kops (Hubrecht Institute, Utrecht, The Netherlands), and human U2OS (osteosarcoma patient) cells with inducible expression of phospho-null CENP-E mutated at the AurA/B-specific phospho-site Threonine 422 (T422A), which was a gift from Marin Barišić (Danish Cancer Institute, Copenhagen, Denmark). All cell lines were cultured in flasks in Dulbecco's Modified Eagle's Medium with 1 g/L D-glucose, pyruvate, and L-glutamine (DMEM, Thermo Fisher, 11995065), supplemented with 10% (vol/vol) heat-inactivated Fetal Bovine Serum (FBS, Sigma-

Aldrich) and penicillin (100 IU/mL)/streptomycin (100 mg/mL) solution (Lonza). The cells were kept at 37 °C and 5% $CO_2$ in a humidified incubator (Galaxy 170 S CO2, Eppendorf) and regularly passaged at the confluence of 70–80%. None of the cell lines were authenticated. All cell lines have also been tested for mycoplasma contamination once a month by examining samples for extracellular DNA staining with SiR-DNA (100 nM, Spirochrome) and Hoechst 33342 dye (1 drop/2 ml of NucBlue Live ReadyProbes Reagent, Thermo Fisher Scientific) and have been confirmed to be mycoplasma free.

### Sample preparation and siRNAs

At 80% confluence, the DMEM was removed from the flask and the cells were washed with 5 ml of phosphate buffered saline (PBS). Then, 1 ml 1% trypsin/ethylenediaminetetraacetic acid (EDTA, Biochrom AG) was added to the flask and cells were incubated at 37 °C and 5% $CO_2$ in a humidified incubator for 5 min. After incubation, trypsin was blocked by adding 4 ml of DMEM. For the RNAi experiments, cells were seeded to reach 60% confluence the next day and cultured on 35 mm uncoated plates with 0.17 mm (#1.5 coverglass) glass thickness (MatTek Corporation) in 1 ml of DMEM with the supplements described above. After one day of growth, cells were transfected with either targeting or non-targeting siRNA constructs which were diluted in OPTI-MEM (Thermo-Fisher) to a final concentration of 100 nM in the medium with cells. All transfections were performed 48 h before imaging using Lipofectamine RNAiMAX Reagent (Life Technologies) according to the instructions provided by the manufacturer, unless otherwise indicated. Codepletions of Hec1 and CENP-E in HeLa cells were performed two times in two subsequent days by the same protocol. After 4 h of siRNA treatment, the medium was changed to the prewarmed DMEM. GFP-Hec1-9A construct in HeLa cells was expressed by addition of 1 µg ml$^{-1}$ doxycycline for 24 h prior to fixation. Proteasome inhibitor MG-132 (Merck, M7449, IC50 value 100 nM) at a final concentration of 1 µM, was added 5 h before fixation for experiments involving HeLa GFP-Hec1-9A cells. Synchronization of GFP-Hec1-9A-expressing cells was achieved by treating cells seeded at 60% confluency with 2 mM thymidine (Sigma) for 24 h, followed by four washes with pre-warmed DMEM. Cells were then incubated with 5 µM RO-3306 (MedChemExpress) for 16 h and subsequently washed three times for 5 minutes each at 37 °C with pre-warmed DMEM. After release, cells were incubated for 2 h in fresh medium, then treated with 1 µM MG-132 (Merck) for 2 h prior to fixation.

The siRNA constructs used were: human CENP-E ON-TARGETplus SMART pool siRNA (L-003252-00-0010, human custom-made 3'UTR HEC1 siRNA (oligo #3, Thermo Fisher Scientific; sequence 5'-CCCUGGGUCGUGUCAGGAA-3'), and control siRNA (D-001810-10-05, Dharmacon). For experiments in U2OS cells expressing different CENP-E variants, endogenous CENP-E was depleted by transfecting cells with 20 nM 3'UTR-targeting siRNA (5'-CCACUAGAGUUGAAA-GAUA-3') 24 h prior to fixation[23]. GFP-CENP-E expression was induced by adding doxycycline (1 µg/ml, Sigma-Aldrich) overnight. 1 µM MG-132 (Merck) was added for 2 h in fresh medium before the fixation.

### Drug treatments and drug washouts

CENP-E inhibitor GSK-923295 (MedChemExpress, IC50 value 3.2 nM) at a final concentration of 80 nM for RPE-1 and U2OS cells, was added 1–4 h before imaging, or when noted, acutely before imaging. Eg5 inhibitor Monastrol (HY-101071A/CS-6183, MedChemExpress, IC$_{50}$ value 50 µM) at a final concentration of 100 µM, was added 3 h before imaging or acutely before imaging when noted. Aurora A inhibitor MLN8054 (MedChemExpress, IC50 value 4nM) at a final concentration of 125 nM or 250 nM, as noted, was added acutely before imaging or 30-60 minutes before fixation. Aurora A inhibitor Alisertib (MLN8237, MedChemExpress, IC$_{50}$ value 1.2 nM) at a final concentration of 125 nM, was added 30 minutes before fixation. Aurora A inhibitor TCS7010 (MedChemExpress, IC$_{50}$ value 3.4 nM) at a final concentration of 100 nM, was added 30 minutes before fixation. Aurora A and B inhibitor ZM-447439 (MedChemExpress, IC$_{50}$ value 130 nM) at a final concentration of 3 µM, was added 15 minutes before fixation. Plk4 inhibitor Centrinone (MedChemExpress, K$_i$ value 0.16 nM) at a final concentration of 300 nM, was added 1-5 days before the imaging as noted in the manuscript. The stock solutions for all drugs were prepared in DMSO. The stock solutions of all drugs were kept aliquoted at 10-50 µL at -20 °C for a maximum period of three months or at -80 °C for a maximum period of six months. New aliquots were thawed weekly for each new experiment. All drugs were added to DMEM used for cell culture to obtain the final concentration of a drug as described. Drug washouts were performed by replacing drug-containing medium with 2 mL of fresh pre-warmed DMEM, followed by four subsequent washouts with 2 mL of pre-warmed DMEM.

To deplete centrioles, Plk4 kinase activity was inhibited with centrinone, a small molecule inhibitor that blocks the centriole duplication cycle, and varied the duration of treatment to obtain spindles with different numbers of centrioles on each centrosome, a method established previously[19]. Subsequently, we treated Plk4-inhibited cells with CENP-E inhibitor or DMSO and imaged cells a few hours after treatment using the same live cell imaging protocol described above. After acute treatment of CENP-E inhibited cells with the Eg5 inhibitor Monastrol (50 µM), spindles quickly shortened, as expected from previous reports[69]. Spindle shortening was also observed in most CENP-E inhibited cells after acute addition of the Aurora A inhibitors MLN8054 (125 nM) and TCS7010 (100 nM), consistent with previous studies[50]. Alisertib (125 nM) did not produce a comparable effect, likely due to the short treatment duration, limited to a maximum of 30 minutes, to accommodate the requirements of our assay. This brief incubation was necessary to prevent the re-congression of polar chromosomes, a phenomenon observed in untreated cells over time. Given that Alisertib is a well-characterized and potent Aurora A inhibitor[50], we suspect that a longer exposure or higher concentration might yield spindle shortening effects similar to those observed with MLN8054 and TCS7010.

Regarding cells imaged by confocal microscopy after the chronic inhibition of CENP-E by GSK-923295 (80 nM), all pseudo-metaphase spindles chosen for imaging were phenotypically similar between each other and between conditions where CENP-E was depleted or reactivated after washout of the CENP-E inhibitor: 1) in all cells few chromosomes were localized close to one of the spindle poles, called polar chromosomes, and their numbers ranged from 2-16 per cell; 2) occasionally in some cells few chromosomes were localized already in between spindle pole and equator; and 3) in all cells most chromosomes were already aligned and tightly packed at the metaphase plate. This type of chromosome arrangement was expected from previously published data that reported that only 10-30% of chromosomes upon NEBD require CENP-E-mediated alignment[8]. Likewise, under all conditions where CENP-E was perturbed and cells imaged by confocal microscopy during pseudo-metaphase, the overall displacement of spindle poles from each other was negligible, consistent with cells being in a pseudo-metaphase state where there is no net change in spindle length. For LLSM-based assay, no inclusion criteria for imaging were used as the cells were non-synchronized and entered mitosis stochastically. Only randomly selected cells that entered mitosis and subsequently entered anaphase during imaging time were used for kinetochore and centrosome tracking analysis.

### Immunofluorescence

For cell labeling with phospho-Knl1, phospho-Dsn1, phospho-Hec1, phospho-Aurora A, phospho-Aurora B, phospho-AuroraA/B/C, RPE-1 cells expressing CENP-A-GFP and centrin1-GFP grown on glass-bottom dishes (14 mm, No. 1.5, MatTek Corporation) were pre-extracted with pre-warmed PEM buffer at 37 °C (0.1 M PIPES, 0.001 M $MgCl_2 \times 6\ H_2O$,

0.001 M EDTA, 0.5% Triton-X-100) for 30 s at room temperature and fixed by 1 ml of pre-warmed solution of 4% paraformaldehyde (PFA) for 20 min. All other antibodies were used on cells fixed for 2 minutes with ice-cold methanol, except for those used in STED microscopy, which are described below. After fixation, cells were washed 3 times for 5 min with 1 ml of PBS and permeabilized with 0.5% Triton-X-100 solution in water for 30 min at room temperature. To block unspecific binding, cells were incubated in 1 ml of blocking buffer (2% bovine serum albumin, BSA) for 2 h at room temperature. The cells were then washed 3 times for 5 min with 1 ml of PBS and incubated with 500 µL of primary antibody solution overnight at 4 °C. Antibody incubation was performed using a blocking solution composed of 0.1% Triton, 1% BSA in PBS.

Following primary antibodies were used: rabbit monoclonal phospho-Knl1 (1.5 mg/ml, gift from J. Welburn, diluted 1:1000), rabbit monoclonal phospho-Dsn1 conjugated to Cy3 (0.5 mg/ml, gift from J. Welburn, diluted 1:250, diluted 1:500), rabbit monoclonal phospho-Aurora A (Thr288) (C39D8) (3079 T, Cell Signaling Technology, diluted 1:500), rabbit monoclonal phospho-Aurora A (Thr288)/Aurora B (Thr232)/Aurora C (Thr198) (D12A11) (2914 T, Cell Signaling Technology, diluted 1:500), rabbit polyclonal phospho-Hec1 (Ser55) (GTX70017, GeneTex, diluted 1:500), rabbit polyclonal phospho-Aurora B (Thr232) (GTX85607, GeneTex, diluted 1:500), rabbit monoclonal Anti-NDC80 (HPA066330-100uL, Sigma-Aldrich, diluted 1:250), human anti-centromere (CREST) protein (15-234, Antibodies Incorporated, 1:500), and rat anti-alpha-tubulin YL1/2 (MA1-80017, Invitrogen, diluted 1:500). Where indicated, DAPI (1 µg/mL) (D9542, Sigma-Aldrich) was used for chromosome visualization.

After primary antibody, cells were washed in PBS and then incubated in 500 µL of secondary antibody solution for 1 h at room temperature. Following secondary antibodies were used: donkey anti-rabbit IgG Alexa Fluor 647 (ab150075, Abcam, diluted 1:1000) for phospho-Dsn1, donkey anti-rabbit IgG Alexa Fluor 594 (ab150064, Abcam, diluted 1:1000) for all other rabbit antibodies, donkey anti-mouse IgG Alexa Fluor 594 (ab150108, Abcam, diluted, 1:1000) for all mouse antibodies, goat anti-human DyLight 594 (Abcam, ab96909, diluted 1:1000), and donkey anti-rat IgG Alexa Fluor 647 (ab150155, Abcam, diluted 1:500). Finally, cells were washed with 1 ml of PBS, 3 times for 10 min. Cells were imaged either immediately following the imaging or were kept at 4 °C before imaging for a maximum period of one week.

To visualize alpha-tubulin in super-resolution by STED and Airyscan microscopy in the RPE-1 CENP-A-GFP centrin1-GFP cell line, the ice-cold methanol protocol was avoided because it destroyed the unstable fraction of microtubules[70]. Instead, cells were washed with cell extraction buffer (CEB) and fixed with 3.2% paraformaldehyde (PFA) and 0.1% glutaraldehyde (GA) in PEM buffer (0.1 M PIPES, 0.001 M MgCl$_2$ × 6 H$_2$O, 0.001 M EGTA, 0.5% Triton-X-100) for 10 min at room temperature. After fixation with PFA and GA, for quenching, cells were incubated in 1 ml of freshly prepared 0.1% borohydride in PBS for 7 min and then in 1 mL of 100 mM NH$_4$Cl and 100 mM glycine in PBS for 10 min at room temperature. To block nonspecific binding of antibodies, cells were incubated in 500 µL blocking/permeabilization buffer (2% normal goat serum and 0.5% Triton-X-100 in water) for 2 h at room temperature. Cells were then incubated in 500 µL of primary antibody solution containing rat anti-alpha-tubulin YL1/2 (MA1-80017, Invitrogen, diluted 1:300) overnight at 4 °C. After incubation with a primary antibody, cells were washed 3 times for 10 min with 1 ml of PBS and then incubated with 500 µl of secondary antibody containing donkey anti-rat IgG Alexa Fluor 594 (ab150156, Abcam, diluted 1:300) for 2 h at room temperature, followed by wash with 1 ml of PBS 3 times for 10 min.

## Imaging

The STED confocal microscope system (Abberior Instruments) and the LSM800 laser scanning confocal system (Zeiss) were used for the live-cell experiments we termed the confocal-based imaging assay. STED microscopy was also used to image all fixed cells in super-resolution, by previously described methods[70,71]. Briefly, STED microscopy was performed using an Expert Line easy3D STED microscope system (Abberior Instruments) with the 100x/1.4NA UPLSAPO100x oil objective (Olympus) and an avalanche photodiode detector (APD). The 488 nm line was used for excitation, with the addition of the 561 nm line for excitation and the 775 nm laser line for depletion during STED super-resolution imaging. The images were acquired using the Imspector software. The xy pixel size for fixed cells was 20 nm, and 10 focal planes were acquired with a 300 nm distance between the planes. For confocal live cell imaging of cells, the xy pixel size was 80 nm and 16 focal images were acquired, with 0.5 µm distance between the planes and 1 min time intervals between different frames. During imaging, live cells were kept at 37 °C and at 5% CO$_2$ in the Okolab stage incubation chamber system (Okolab). Live-cell imaging following acute reactivation or depletion of CENP-E (Supplementary Fig. 1) was performed using an Opterra I point-scanning confocal system (Bruker), as previously described[11].

LSM 800 confocal laser scanning microscope system (Zeiss) was used for the rest of fixed-cell confocal-based imaging in hTERT-RPE-1 cells expressing CENP-A-GFP and centrin1-GFP with the following parameters: sampling in xy, 0.27 µm; z step size, 0.5 µm; total number of slices, 32; pinhole, 48.9 µm; unidirectional scan speed, 10; averaging, 2; 63x Oil DIC f/ELYRA objective (1.4 NA), 488 nm laser line (0.1–1% power for different experiments), and detection ranges of 450–558 nm for the green channel, 561 nm laser line (0.1–1% power for different experiments) and detection range of 565–650 nm for the red channel, 640 nm laser line (0.1–1% power for different experiments), and detection range of 656–700 nm for the far red channel, and 405 nm laser line (0.5% power) and detection range of 400–450 for the blue channel. Images were acquired using ZEN 2.6 and later versions (blue edition; Zeiss). Imaging was done on multiple positions determined by the user simultaneously, and at the confocal lateral resolution. For imaging of tubulin in super-resolution following acute Aurora A inhibition in CENP-E-inhibited cells, Airyscan mode of LSM 800 was used. Following parameters were used: sampling in xy, 2 µm; z step size, 0.146 µm; total number of slices, 41; pinhole, 265 µm; unidirectional scan speed, 7; averaging, 2; 63x Oil DIC f/ELYRA objective (1.4 NA), 488 nm laser line (1.5% power), 561 nm laser line (1% power) and detection ranges of 450–580 and 565–700, for the green and red channels, respectively. Imaging was done on multiple positions determined by the user.

The Lattice Lightsheet 7 microscope system (Zeiss) was used for live cell imaging of hTERT-RPE-1 cells expressing CENP-A-GFP and centrin1-GFP in an assay we termed the LLSM-based imaging assay. The system was equipped with an illumination objective lens 13.3×/0.4 NA (at a 30° angle to cover the glass) with a static phase element and a detection objective lens 44.83×/1.0 NA (at a 60° angle to cover the glass) with an Alvarez manipulator. Images were acquired using ZEN 2.7 and later versions (blue edition; Zeiss). The automatic immersion of water was applied from the motorized dispenser at an interval of 20 or 30 minutes. Right after sample mounting, three steps of the 'create immersion' auto-immersion option were applied. The sample was illuminated with a 488-nm diode laser (power output 10 mW) with laser power set to 1-2%. The detection module consisted of a Hamamatsu ORCA-Fusion sCMOS camera with exposure time set to 15-20 ms. The LBF 405/488/561/642 emission filter was used. During imaging, cells were kept at 37 °C and at 5% CO$_2$ in a Zeiss stage incubation chamber system (Zeiss). The imaging area's width in the x-dimension ranged from 1 to 1.5 mm, with a 0.4 µm interval size. The time between consecutive frames varied from 30 seconds to 1 minute, depending on the chosen imaging area width. The total imaging duration, set for 1 to 1.5 days, was occasionally interrupted by air bubbles, which caused a loss of intensity in part or all of the imaging area. When the entire area

was affected by air bubbles, the image was cropped in ZEN software to reduce the final file size before further processing. Some movies, which were later processed for color-coding, were deskewed using ZEN 3.7 software with the "Linear Interpolation" and "Cover Glass Transformation" settings. The light sheet's length, also referred to as the field of view or illumination width, was 30 μm, while its thickness was set to 1000 nm. The parameters 'Focus sheet,' 'Focus Waist,' and 'Aberration Control' were manually fine-tuned before each imaging session, with ranges of -0.170 to -0.230, 50 to 60, and 170 to 185, respectively.

### Tracking of centrosomes and kinetochores

The spatial x and y coordinates of the kinetochore pairs were extracted in each time frame using the Low Light Tracking Tool (v.0.10), an ImageJ plugin based on the Nested Maximum Likelihood Algorithm (https://imagej.net/plugins/low-light-tracking-tool), as previously described[69]. Tracking of polar kinetochores in the x and y planes was performed on the maximum intensity projections of all acquired z planes. Some kinetochore pairs could not be successfully tracked in all frames, mainly owing to cell and spindle movements in the z-direction over time. Spindle poles were manually tracked with points placed between the center of the two centrioles or centriole in centrinone-treated cells. Kinetochore pairs that were aligned at the start of the imaging in a confocal-based imaging assay were also manually tracked in two dimensions. Quantitative analysis of all parameters was performed using custom-made MATLAB scripts (MatlabR2021a 9.10.0). Due to the inability to reliably track polar kinetochores across all time frames due to neighboring kinetochores, tracking of polar kinetochores routinely commenced a few frames before the kinetochore pair began moving towards the equatorial plane. In the CENP-E reactivated and CENP-E-depleted RPE-1 cells imaged by confocal microscopy, kinetochores were tracked from the onset of imaging until the successful identification of the same kinetochore pair, which occurred when the kinetochore pair was lost in the imaging plane, among other kinetochores, or when the imaging session ended, whichever came first.

### Quantification of mean pHec1, pKnl1 and pAuroraB intensities on kinetochore pairs and pAuroraA on spindle poles

The fluorescence intensity signal of each kinetochore in both CENP-A and respective kinetochore protein channel (pDsn1, pKnl1) was measured by using the "Oval selection" tool in ImageJ with the size and position defined by the borders of the CENP-A signal of each kinetochore in the sum intensity projection of all acquired z-planes. pAurora A signal was measured using the "Oval selection" tool around the centriole signal as the center or around the signal of the Aurora A of the same size on the pole without centriole. The background fluorescence intensity measured in the cytoplasm was subtracted from the obtained values, and the calculated integrated intensity value was divided by the number of z-stacks used to generate the sum projection of each cell. The obtained mean intensity value subtracted for background for each kinetochore was normalized to the intensity of the CENP-A signal. For the pHec1 labeling experiments, the mean signal of Hec1 and CENP-A was measured by the same approach but on one z-plane where kinetochore CENP-A signal was the highest. The mean signal of the pAurora B was measured between extent of CENP-A signals of sister kinetochore pairs, and the mean value was normalized to mean value of CENP-A signal of each sister kinetochore pair. All movies used for the analysis were obtained after using the same fixation protocol and were imaged with the same imaging parameters in at least three independent biological replicates. To avoid measurement of non-specific labeling of the spindle poles that are known artefacts of phospho-antibodies to kinetochore proteins (pAurB, pKnl1, pHec1)[30], only the kinetochores outside of the circular pole signal (~1 μm from the centrosome) were measured and used for analysis.

### Quantification of mean signal intensities of proteins after RNAi interreference

The fluorescence intensity signals were measured in triplicate samples for targeting and non-targeting groups for each siRNA treatment. Each siRNA targeting and non-targeting triplicates were imaged by the same protocol. Proteins were measured in mitotic cells at their respective locations during mitosis by using the "Polygon selection tool", as established by many previous studies: CENP-E and Hec1 were measured in area occupied by chromosomes, as defined by the DAPI on sum of all acquired planes.

### Quantification of polar kinetochore pairs

In all experiments, polar kinetochores were defined as kinetochore pairs positioned closer to one of the spindle poles than to the center of the equatorial plane, with all others classified as aligned. In fixed-cell images, the polar category included both kinetochores near the pole and those in transit toward the metaphase plate. Kinetochore pairs with at least one kinetochore located 3 μm or less from the equatorial plane were assigned to the aligned group. In confocal-based experiments, polar chromosomes were defined from the onset of imaging. In cells treated with centrinone and their respective DMSO controls, the number of polar chromosomes was measured at the time when the spindle finished the bipolarization process, e.g., prometaphase elongation of the spindle. This was done because all spindles of the 1:0 type with only one centriole at the beginning of mitosis went into the monopolar state and stayed in this state for variable durations before bipolarization, ranging from 10 to 50 minutes, as reported previously[26]. The residence time of polar chromosomes on spindle poles, i.e., the efficiency of congression for each pole, was defined as the time from spindle bipolarization until the polar chromosome either entered the metaphase plate or the cell entered anaphase with an uncongressed polar chromosome. Duration of mitosis was defined as the time from the first signs of nuclear envelope breakdown to one frame before the visible onset of anaphase, defined by the separation of sister kinetochore pairs. In fixed cells, the number of polar chromosomes per single cell, as defined above, was used as a readout of congression efficiency.

Congression velocity was calculated for each pair of kinetochores in the last 6 minutes before the center of the pair of kinetochores surpassed 2 μm from the equatorial plane measured as the nearest distance from a center of a pair to a plane. These times represent the fast phase of kinetochore movement towards the plate[11]. An aligned kinetochore pair was defined as every pair that was found 3 μm from the equatorial plane at any given time. The equatorial or metaphase plane was defined in each time frame as the line perpendicular to the line connecting the centrosomes at their midpoint. The angle between the kinetochore pairs and the main spindle axis was defined as an angle between a line connecting two centrosomes and a line connecting the center of the kinetochore pair and a spindle pole nearest to the respective pair. The angle between the midpoint of sister kinetochores and the centrosome-to-centrosome axis was measured using the angle tool in ImageJ. The successful alignment of a polar kinetochore pair to the metaphase plate was defined as the moment when the midpoint between sister kinetochores first crossed within 2μm of the equatorial plane.

Maximum metaphase plate spread was measured in ImageJ as the distance between the outermost chromosome boundaries along the main spindle axis, based on the DAPI signal. The number of polar chromosomes was determined by counting kinetochore pairs in CREST-labeled cells or individual chromosomes in DAPI-labeled cells. In Hec1-9A overexpression experiments, no selection bias was applied for Hec1-9A–expressing cells; however, the proportion of expressing cells was consistent across experiments, accounting for approximately 10% of imaged cells. Similarly, in CENP-E-T422 overexpression experiments, no selection bias was applied, and CENP-E-T422–expressing cells represented less than ~5% of

imaged cells. Multipolar spindles, identified by tubulin or centrin staining, were excluded from analysis. Spindle length was measured in ImageJ as the distance between centriole pairs in 2D maximum projections. The distance to the nearest centrosome or to the equatorial plane was measured from the midpoint between sister kinetochores to either the center of the nearest centriole pair or the equatorial plane, respectively. Interkinetochore distance was measured as the distance between the centers of two sister kinetochores.

## Image processing and statistical analysis

Image processing was performed in ImageJ (National Institutes of Health). Quantification and statistical analysis were performed in MatLab. The figures were assembled in Adobe Illustrator (Adobe Systems). Raw images were used for quantification. The images of spindles were rotated in every frame to fit the long axis of the spindle to be parallel with the central long axis of the box in ImageJ and the spindle short axis to be parallel with the central short axis of the designated box in ImageJ. The designated box sizes were cut to the same dimensions for all panels in the figures where the same experimental setups were used across the treatments. When comparing different treatments in channels in which the same protein was labeled, the minimum and maximum intensity of that channel was set to the values in the control treatment for all figures. When indicated, the smoothing of the images was performed using the "Gaussian blur" function in ImageJ (s = 0.5-1.0). Color-coded maximum intensity projections of the z-stacks were done using the "*Temporal color code*" tool in Fiji by applying "16-color" or "Spectrum" lookup-table (LUT) or other LUT as indicated. For the generation of univariate scatter plots, the open Matlab extension "UnivarScatter" was used (https://github.com/manulera/UnivarScatter).

Data are given as mea$n \pm$ standard deviation (st.d.), unless otherwise stated. The mean line was plotted to encompass a minimum of 60% of the data points for each treatment. Other dispersion measures used are defined in their respective figure captions. The exact values of n are given in the respective figure captions, where n represents the number of cells or the number of tracked kinetochore pairs, as defined for each n in the figure captions or tables. The number of independent biological replicates is also given in the figure captions. An independent experiment for acute drug treatments was defined by the separate addition of a drug to a population of cells in an independently prepared dish. The number of cells imaged simultaneously by confocal microscopy ranged from 1 to 7, depending on the specific microscopy system used. The $p$ values when comparing data from multiple classes that followed a normal distribution were obtained using the one-way analysis of variance (ANOVA) test followed by pairwise Two-sided Tukey's Honest Significant Difference (HSD) test (significance level was 5%). $p < 0.05$ was considered statistically significant, very significant if $0.001 < p < 0.01$ and extremely significant if $p < 0.001$. Values of all significant differences are given with the degree of significance indicated ($*0.01 < p < 0.05$, $**0.001 < p < 0.01$, $***p < 0.001$, $**** < 0.0001$). Linear regression was performed using ordinary least squares. The significance of the slope was assessed with a two-tailed t-test, and goodness-of-fit was evaluated using $R^2$. The exact p-values are given in the Source data file[72]. No statistical methods were used to predetermine the sample size. The experiments were not randomized and, except where stated, the investigators were not blinded to allocation during experiments and outcome evaluation.

## Reporting summary

Further information on research design is available in the Nature Portfolio Reporting Summary linked to this article.

## Data availability

The source data used for the main figures have been deposited in the Figshare database (https://doi.org/10.6084/m9.figshare.29135756)[72]. All other data supporting the findings of this study are available, and access can be obtained from the corresponding authors.

## Code availability

Codes used to analyze and plot the data are available from the corresponding authors on request.

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

## Acknowledgements

We thank Alexey Khodjakov, Geert Kops and Marin Barišić for cell lines; Marin Barišić, Carlos Conde, Helder Maiato, and Andrea Musacchio for reviewing and discussing the results; Julie Welburn for discussions and antibodies; Ivana Šarić for the drawings; and members of the Tolić group and Nenad Pavin group for constructive comments on the manuscript. This work was funded by the European Research Council (ERC Synergy Grant, GA Number 855158), the Croatian Science Foundation (HRZZ) through Swiss-Croatian Bilateral Projects (project IPCH-2022-10-9344), and projects co-financed by the Croatian Government and the European Union through the European Regional Development Fund—the Competitiveness and Cohesion Operational Programme: IPSted (Grant KK.01.1.1.04.0057) and QuantiXLie Center of Excellence (Grant KK.01.1.1.01.0004). This research was performed using services, storage and computing resources provided by the University of Zagreb University Computing Centre – SRCE.

## Author contributions

K.V. and I.M.T. conceived the project, K.V. performed all experiments, quantified, analyzed, and presented the data, K.V. conceptualized and prepared the original draft, K.V. and I.M.T. reviewed, edited, and discussed the manuscript.

## Competing interests

The authors declare no competing interests.
