## [Transparent Peer Review file · Nature Communications]

Kinetochores-centrosome feedback linking CENP-E and Aurora kinases controls chromosome congression

Corresponding Author: Professor Iva Tolić

Version 0:

Reviewer comments:

Reviewer #1

(Remarks to the Author)

Chromosome movement toward the spindle equator is crucial for accurate cell division. Mitotic kinesin CENP-E is required for a prompt congression of chromosomes to the equator during prometaphase-metaphase transition. Current models argue that CENP-E drives congression by gliding kinetochores along microtubules independently of chromosome biorientation. Vukušić and Tolić now present emerging evidence in which they propose a biochemical cascade of Aurora A-Aurora B-CENP-E signaling in guiding chromosome congression. The cellular characterization is well-performed but exclusively rely on chemical inhibitors that possess dose-dependent off-target effects. Although this reviewer favors prompt publishing this set of experimental outcomes, following points should be fully addressed in revised manuscript before consideration for publication.

Major Points:

- 1, Most of functional analyses were based on a combination of siRNA and chemical inhibitors of Aurora kinases which do not give sufficient time resolution to delineate mitotic chromosome movements discussed in this study. I would recommend authors to employ dTAG or AID system to perform time-resolved degradation of CENP-E to validate the chemical inhibitor experiments.
2. Fig. 3D is a hypothetical model without firm evidence such as tubulin labeling. In fact, it would be strong evidence if transmission electron microscopic analysis is performed under this condition.
3. Fig. 4A should split all channels and add ACA staining as a baseline for Hec1 signal normalization. In fact, it would be stronger if tubulin staining can be included in Fig. 4 to demonstrate the microtubule connection.
4. A few highly relevant references on end-on capture of spindle microtubules should be included (PMID: 23891108, 28751710, 31201382)

Reviewer #2

(Remarks to the Author)

The manuscript presents an investigation into the role of centrosomes in regulating chromosome congression under varying levels of CENP-E activity. The study integrates multiple experimental approaches, including RNAi, chemical inhibition, and advanced live-cell imaging, to elucidate how centrosomes and Aurora kinases influence kinetochore function. The findings propose a feedback mechanism where centrosomes, through Aurora A activity, regulate Aurora B at kinetochores to control chromosome congression timing.

The manuscript provides compelling evidence that centrosomes inhibit chromosome congression in the absence of CENP-E. It establishes that high Aurora A activity at centriolar poles enhances Aurora B activity at kinetochores, leading to increased phosphorylation of microtubule-binding proteins and delayed congression. The study successfully identifies a feedback loop between CENP-E and Aurora kinases, which modulates kinetochore-microtubule interactions. These findings contribute significantly to the understanding of chromosome congression and spindle dynamics.

The proposed model expands the understanding of chromosome congression beyond the conventional role of CENP-E as a motor protein. The study highlights a previously underappreciated inhibitory role of Aurora A in congression, which could

have implications for cancer biology, particularly in the context of Aurora kinase overexpression. Given the importance of chromosome congression for accurate mitotic progression, these findings are highly relevant to cell biology and cancer research.

The following points should be addressed for improvement:

- 1) The working model is proposed heavily on data from experiments in which CENP-E was depleted or chemically inhibited. However, in unperturbed mitosis, CENP-E is usually active in either non-transformed cells or cancer cells. The authors should discuss the implication of their conclusion in normal mitosis progression.
- 2) The author claimed the direct regulation of Aurora B by Aurora A. Direct biochemical assays demonstrating a physical or functional interaction between these two kinases would further substantiate the proposed model.
- 3) The discussion appropriately contextualizes the results, but alternative explanations for certain findings (such as potential indirect effects of Plk4 inhibition) should be considered.
- 4) The criteria for defining congression efficiency, as well as the statistical methods used for quantification, should be more explicitly described.
- 5) The discussion should address potential alternative mechanisms for the observed effects, particularly the role of additional spindle-associated proteins in congression regulation.
- 6) I would suggest the authors to provide additional experiments to address why the Hec1-9D mutant does not rescue CENP-E inhibition in the context of Hec1 depletion.
- 7) Where necessary, I would suggest the authors to include additional controls or validation to address the potential for non-specific binding of phospho-specific antibodies.

Reviewer #3

(Remarks to the Author)

In the work "Kinetochore-centrosome feedback linking CENP-E and Aurora kinases controls chromosome congression" the authors suggest a feedback loop involving Aurora Kinases and CENP-E. The novelty of the work lies in highlighting that centrosomes primarily inhibit congression initiation when CENP-E is inactive or absent. They have achieved this by removing centrioles using a PLK4 inhibitor, centrinone. This treatment allows chromosomes near acentriolar poles to initiate congression independently of CENP-E. Kinetochores near centriole bearing poles with high Aurora A activity further enhances Aurora B activity - this is thought to increase phosphorylation of microtubule-binding proteins at kinetochores and preventing stable microtubule attachments in the absence of CENP-E. This is rescued by inhibiting Aurora A as they find reduction in Aurora B activity, initiating congression without the CENP-E motor. Overall this is a good advance for the chromosome segregation field and enough detail is provided for the methods of the work to be reproduced.

Major comments

Figure 1: This is the strongest figure in the paper that shows the difficulty in congression near centrioles but not away – it would help if they could reproduce it in STLC-released cells to avoid any bias from end-point phenotypes (what if centriole bearing pole were dominant in the initial capture of chromosomes, then again one wouldn't see chromosomes at the centriole lacking pole). Why is figure 1C (Control) image blurrier compared to inhibitor and depleted conditions? Is this reflecting movements?

Figure 2 shows that Centriolar Aurora A activity is linked to increased activity of kinetochore Aurora B and its downstream KMN network targets without CENP-E activity. This is expected since Aurora A and B are known to have overlapping substrate recognition sites. However, this does not detract from the significance of their findings in this paper. Lack of centrioles allows the authors to elegantly establish the presence of excessive phosphoKNL1 in these conditions.

Figure 3 shows that Aurora-A directly upregulates Aurora-B using polar and nonpolar centromeres. This topic and conclusion have been reviewed before: PMC10040841. Can the authors help clarify how their work differs from past findings. If the difference is only the tool then Figures 2 and 3 can be easily combined and pAurB linked to the model. Have the authors tried other AuroraA inhibitors? MLN8054? or Aurora-A depletions. An alternate tool can help this part of the study to be stronger.

Figure 4: This is probably the weakest figure of the study.

Double depletions are hard to control. So it's ideal to assess the HEC1 and CENP-E protein levels independently in all conditions either with IF or IB. Frequently partially depleted kinetochores would result in congressed kinetochores while fully depleted ones would fail to align completely.

Also, uncongressed chromosome measurements, as shown in Figure-4, must be combined with pole-to-pole axis.

Especially since some of the treatments are also known to tumble the spindle position (for example the one presented in the leftmost panel has the entire metaphase rosette (and empty centre) visible compared to the HEC1 depletion. (middle panel). Since this is the main conclusion of the work, I would suggest measuring centromeric positions relative to spindle pole positions to confirm the outcomes.

Minor comments

1. Do 1:1, 1:0 and 0:0 differ in their mitotic lengths or IK distances?
2. In figure 3, P and A are clear? What happens to those on transit? Do they exclude them or assign them to either of the categories. They may have different behaviour- this is particularly important for pHEC1.
3. Please clarify the sentence "This effect, however, did not occur after CENP-E inhibition (Fig. 4C)."
4. Minor points on discussion:

It is unclear why the authors do not conclude that kinetochore-centrosome feedback is between aurora kinases and the state of 'attachment'. CENP-E loss is known to affect the status of attachment (lateral or not) which will broadly affect a variety of

outer-kinetochore protein status and feedback loop.

5. Would this feedback loop be present or absent in oocytes which naturally lack centrioles and have large spindle pole to kinetochore distances?

6. Did they observe an increase in syntelic versus monotelic or 'behind the pole kinetochores' in their centrinone treatment study? Would polar chromosomes require centrioles to be captured normally?

Version 1:

Reviewer comments:

Reviewer #1

(Remarks to the Author)

Most of my concerns have been properly addressed. I think the manuscript is now suitable for publication.

Reviewer #2

(Remarks to the Author)

In the revised manuscript, the authors have addressed mostly of my concerns, through either carrying out additional experiments or extending the discussion. I therefore suggest that the revised manuscript is acceptable for publication.

Reviewer #3

(Remarks to the Author)

In NCOMMS-24-76220A, the authors present evidence for a kinetochore-centrosome feedback linking CENP-E and Aurora kinases's role in chromosome congression. Their localised model for regulating chromosome congression is stronger with new experiments – including additional inhibitors, HEC1 and CENPE mutants.

Below are minor queries for clarity and accuracy.

Figure 2. It would help to clarify the conclusion on 'spatial bias': "These results support a model in which CENP-E-dependent congression of polar chromosomes is spatially biased toward centriolar spindle poles, both in non-transformed and transformed cells and across distinct modes of CENP-E perturbation (Fig. 2d)".

Line 18: what does 'self-limiting' mean? Please describe in results or explain in discussion.

Line 117. This sentence requires clarification. "These findings suggest that centrioles specifically limit the initiation of polar chromosome congression when CENP-E activity is compromised." Clarifying this sentence is important because of a preceding sentence in 108 states "The acentriolar poles (labeled "0") in 1:0 spindles consistently had an average of around one polar chromosome, regardless of CENP-E activity".

Line 223 Typo in inhibitor name: "inhibition by 3 μ M ZM-4473 and"

Line 337 "all in the context of endogenous Hec1 depletion" this work needs some evidence for endogenous Hec1 depletion. (Eg., cold treatment, endogenous Hec1 levels or an immunoblot to showcase loss of endogenous Hec1 as it's a notoriously difficult protein to be fully depleted from the kinetochore).

Line 375: Please double-check kinetochore numbers and include for all conditions if required "N_u_m_b_e_r_s_: 5_7_1_k_i_n_e_t_o_c_h_o_r_e_p_a_i_r_s_f_r_o_m_9_1_c_e_l_l_s_(c), 3_2_7_c_e_l_l_s_(d)_2_0_3_c_e_l_l_s_(e), a_l_l_f_r_o_m_≥3_r_e_p_l_i_c_a_t_e_s"

Line 578: typo 'final final' "IC50 value 3.4 nM) at a final final concentration of 100 nM,"

Multiple Aurora inhibitors are being named – some but not caused a phenotype. Please discuss the differences induced by these inhibitors (in results/discussion) as it would help future users.

"Aurora A inhibitor Alisertib (MLN8237, MedChemExpress, IC50 value 1.2 nM) at a final final concentration of 125 nM, was added 30 minutes before fixation. Aurora A inhibitor TCS7010 (MedChemExpress, IC50 value 3.4 nM) at a final final concentration of 100 nM, was added 30 minutes before fixation. Aurora B inhibitor ZM-447439 (MedChemExpress, IC50 value 130 nM) at a final concentration of 3 μ M, was added 15 minutes before fixation."

ZM 447439 can inhibit both Aurora A and B. Please comment or discuss.

Version 2:

Reviewer comments:

Reviewer #3

(Remarks to the Author)

The authors have addressed my queries satisfactorily. Changes to results text and limitations text have made the work stronger.

As it's one of early lattice light sheet manuscript article, I would strongly recommend the authors to indicate details of light

sheet thickness and length (not the imaging length). The details on laser illumination and objectives have all been added in the previous round of revision and these are useful, but light sheet details per se could be added for reproducibility.

Reviewer #1 (Remarks to the Author):

Chromosome movement toward the spindle equator is crucial for accurate cell division. Mitotic kinesin CENP-E is required for a prompt congression of chromosomes to the equator during prometaphase-metaphase transition. Current models argue that CENP-E drives congression by gliding kinetochores along microtubules independently of chromosome biorientation. Vukušić and Tolić now present emerging evidence in which they propose a biochemical cascade of Aurora A-Aurora B-CENP-E signaling in guiding chromosome congression. The cellular characterization is well-performed but exclusively rely on chemical inhibitors that possess dose-dependent off-target effects. Although this reviewer favors prompt publishing this set of experimental outcomes, following points should be fully addressed in revised manuscript before consideration for publication.

Major Points:

1, Most of functional analyses were based on a combination of siRNA and chemical inhibitors of Aurora kinases which do not give sufficient time resolution to delineate mitotic chromosome movements discussed in this study. I would recommend authors to employ dTAG or AID system to perform time-resolved degradation of CENP-E to validate the chemical inhibitor experiments.

The authors: We thank the reviewer for this constructive comment. In our view, the use of CENP-E inhibitors currently represents the fastest and most effective method to acutely perturb CENP-E activity, enabling the investigation of its immediate effects on mitotic chromosome dynamics. In contrast, systems such as the auxin-inducible degron (AID) require tens of minutes to hours for efficient protein depletion.

We agree that an orthogonal approach based on a different technology—particularly one that retains most of the CENP-E protein but selectively perturbs its function—can provide valuable complementary insights. To this end, we employed a stable U2OS cell line with doxycycline-inducible expression of a phospho-null CENP-E mutant in which the Aurora A/B-specific phosphorylation site at threonine 422 is mutated (T422A). This cell line was obtained from Eibes et al., 2024 (10.1038/s41467-023-41091-2). Cells expressing the CENP-E-T422A mutant exhibit chromosome congression defects similar to those observed following CENP-E inhibition or depletion (Kim et al., 2010, *Cell*; Eibes et al., 2024, *Nat Commun*).

Our goal was to determine whether polar chromosomes on acentriolar spindle poles remain dependent on CENP-E function in cells expressing this phospho-null variant, as observed following CENP-E depletion or pharmacological inhibition in RPE-1 cells. Additionally, we sought to assess whether the spatial bias in CENP-E function—specifically its preferential requirement at centriolar spindle poles—also applies to a different cell type, namely osteosarcoma-derived U2OS cells.

We conducted live-cell immunofluorescence microscopy of mitotic spindles in the described U2OS cell line under the following conditions:

1. Depletion of endogenous CENP-E and expression of GFP–CENP-E-T422A
2. Depletion of endogenous CENP-E alone
3. Treatment with a small molecule CENP-E inhibitor

All cells were continuously treated with the Plk4 inhibitor centrinone for two days to generate a mixed population of cells with either one (1:1) or zero (1:0) centrioles per spindle pole. Centromeres (CREST) and centrioles were labeled. We quantified the number of polar chromosomes associated with each spindle pole type, centriolar (1) or acentriolar (0). The new data have been incorporated into the **Results** section.

“To test whether selective disruption of CENP-E activity, without complete depletion or inducing rigor microtubule binding of the motor, would also bias accumulation of polar chromosomes to the centriolar pole, we used osteosarcoma U2OS cells engineered for doxycycline-inducible expression of phospho-null Threonine 422 (T422A) mutant lacking the Aurora A/B-specific phosphorylation site²³. To compare different modes of CENP-E perturbation in transformed cells, we analyzed three conditions: 1) endogenous CENP-E depletion, 2) pharmacological inhibition of CENP-E, 3) and expression of T422A CENP-E following depletion of endogenous CENP-E. Immunofluorescence microscopy was performed under continuous centrinone treatment for 2 days to generate mixed populations of cells with either centriolar (1:1) or mixed centriolar and acentriolar (1:0) spindle poles (Fig. 2b). We quantified the number of polar chromosomes relative to the number of centrioles on the spindle pole. Consistent with our findings in RPE-1 cells (Fig. 1e, f), polar chromosomes predominantly accumulated at centriolar poles under all conditions that impaired CENP-E activity, including expression of the T422A mutant (Fig. 2c). These results support a model in which CENP-E-dependent congression of polar chromosomes is spatially biased toward centriolar spindle poles, both in non-transformed and transformed cells and across distinct modes of CENP-E perturbation (Fig. 2d).”

2. Fig. 3D is a hypothetical model without firm evidence such as tubulin labeling. In fact, it would be strong evidence if transmission electron microscopic analysis is performed under this condition.

The authors: We agree with the reviewer’s suggestion. Although TEM imaging is beyond the scope of our current revision timeline, we performed new super-resolution imaging to validate the formation of end-on attachments. To test our hypothesis that Aurora A inhibition promotes end-on attachment formation leading to chromosome congression, we conducted tubulin staining in cells pre-treated with a CENP-E inhibitor, followed by acute treatment with the Aurora A inhibitor MLN8054. These cells were then imaged using super-resolution microscopy.

As predicted based on our prior results, kinetochores undergoing congression exhibited end-on microtubule attachments. This finding is consistent with our other assays assessing microtubule–kinetochore interactions during chromosome congression (see accompanying manuscript for details). The new data have been incorporated into the **Results** section.

“Our model suggests that downregulation of Aurora kinases promotes chromosome congression by stabilizing end-on attachments at polar kinetochores. To test this, we stained tubulin in cells

pre-treated with a CENP-E inhibitor, followed by acute treatment with the Aurora A inhibitor MLN8054 (Fig. 4h). Super-resolution Airyscan imaging⁴¹ revealed that most kinetochores located between the centrosome and the metaphase plate had end-on microtubule attachments after acute MLN8054 treatment in CENP-E–inhibited cells (n=15 cells) (Fig. 4h, Extended Data Fig. 3g). These results are consistent with our recent findings showing that acute CENP-E reactivation leads to stabilized end-on attachments on congressing chromosomes¹¹. Together, these findings suggest that Aurora A modulates Aurora B activity near spindle poles, reinforcing our model of localized regulation of chromosome congression.”

3. Fig. 4A should split all channels and add ACA staining as a baseline for Hec1 signal normalization. In fact, it would be stronger if tubulin staining can be included in Fig. 4 to demonstrate the microtubule connection.

The authors: We agree with the reviewer’s comment and are grateful for the suggestion. To address this point, we conducted a new set of experiments using HeLa Hec1-9A-GFP cells. Cells were first synchronized in G1 phase and subsequently in G2 phase, during which we depleted endogenous Hec1 and/or CENP-E. This approach ensured that cells did not enter mitosis in the absence of these proteins (as also highlighted by reviewer 3). Following synchronization, cells were released into mitosis and incubated with a proteasome inhibitor for no longer than two hours to arrest them in metaphase while minimizing confounding effects from prolonged mitotic arrest.

We analyzed four experimental conditions and stained for tubulin and CREST in all cases:

1. Depletion of endogenous Hec1 and CENP-E
2. Depletion of endogenous Hec1 with expression of Hec1-9A-GFP
3. Depletion of endogenous Hec1 and CENP-E with expression of Hec1-9A-GFP
4. Depletion of endogenous Hec1 with Hec1-9A-GFP expression and CENP-E inhibition

We quantified both the maximum chromosome spread and the number of polar chromosomes per cell. Our results show that expression of Hec1-9A in cells lacking both endogenous Hec1 and CENP-E significantly improved chromosome alignment, with an approximately two-fold increase in congression efficiency compared to cells without Hec1-9A. This improvement was reflected in both reduced metaphase plate spread and fewer polar chromosomes, in line with our previous observations. Hec1-9A overexpression did not improve congression when CENP-E was pharmacologically inhibited, likely due to hyperexpanded fibrous coronas preventing stable kinetochore–microtubule attachments on polar chromosomes. Supporting this interpretation, we observed no end-on attachments to spindle poles on polar kinetochores, including syntelic configurations, despite robust Hec1-9A expression. These new data have been incorporated into the **Results** section.

“Based on our finding that acute inhibition of Aurora kinases induces chromosome congression independently of CENP-E¹¹, we hypothesized that constitutive dephosphorylation of Hec1 might compensate for disruptions of CENP-E function after depletion of endogenous Hec1. To control for potential mitotic prolongation effects following combined Hec1 and CENP-E

perturbations, we synchronized cells in G1 before depleting endogenous Hec1, CENP-E, or both, thereby ensuring depletion prior to mitotic entry. After release into G2 and mitosis, cells were arrested in metaphase using a proteasome inhibitor for a maximum of 2 hours (Fig. 5a, bottom).

We assessed chromosome congression efficiency by immunostaining cells for tubulin and centromeres under four conditions, all in the context of endogenous Hec1 depletion: and (1) CENP-E depletion, (2) Hec1-9A expression, (3) Hec1-9A expression with CENP-E depletion, (4) Hec1-9A expression with CENP-E inhibition. Expression of Hec1-9A in cells lacking endogenous Hec1 rescued major chromosome congression defects in both synchronized and unsynchronized cells (Fig. 5b, c, e), as previously reported^{43,44}. Remarkably, expression of Hec1-9A also significantly improved chromosome congression in cells depleted of both endogenous Hec1 and CENP-E (Fig. 5b). The improvement was reflected in both the maximum chromosome spread and the average number of polar chromosomes, which closely matched those observed in cells with intact CENP-E, in synchronized (Fig. 5b, e, Extended data Fig. 3j) as well as unsynchronized cells (Fig. 5c, Extended data Fig. 3i). In all conditions where Hec1-9A was overexpressed, aligned chromosomes had a large interkinetochore distance and showed end-on microtubule attachments (Fig. 5b). Surprisingly, Hec1-9A overexpression did not rescue the congression defects caused by CENP-E inhibition in either synchronized or unsynchronized cells (Fig. 5b, c and e). This is likely due to hyperexpansion of the fibrous corona on polar kinetochores following CENP-E inhibition, which interferes with the stabilization of microtubule attachments¹¹. Consistent with this, the location and interkinetochore distance of polar kinetochores in cells overexpressing Hec1-9A under CENP-E inhibition (Fig. 5b) were similar to those under CENP-E inhibition alone (Fig. 4d). These findings suggest that constitutive Hec1 dephosphorylation is sufficient to drive chromosome congression in the absence of CENP-E and endogenous Hec1, but only when excessive fibrous corona expansion is avoided.”

4. A few highly relevant references on end-on capture of spindle microtubules should be included (PMID: 23891108, 28751710, 31201382)

The authors: We have now included citations to those articles in the **Introduction**.

“We propose that chromosome congression is coupled to biorientation near the spindle poles through CENP-E, which stabilizes end-on attachments at polar kinetochores via its interaction with BubR1¹¹. Thus, chromosome congression requires conversion from lateral to end-on attachments¹²⁻¹⁴.”

Reviewer #2 (Remarks to the Author):

The manuscript presents an investigation into the role of centrosomes in regulating chromosome congression under varying levels of CENP-E activity. The study integrates multiple experimental approaches, including RNAi, chemical inhibition, and advanced live-cell imaging, to elucidate how centrosomes and Aurora kinases influence kinetochore function. The findings propose a feedback mechanism where centrosomes, through Aurora A activity, regulate Aurora B at kinetochores to control chromosome congression timing.

The manuscript provides compelling evidence that centrosomes inhibit chromosome congression in the absence of CENP-E. It establishes that high Aurora A activity at centriolar poles enhances Aurora B activity at kinetochores, leading to increased phosphorylation of microtubule-binding proteins and delayed congression. The study successfully identifies a feedback loop between CENP-E and Aurora kinases, which modulates kinetochore-microtubule interactions. These findings contribute significantly to the understanding of chromosome congression and spindle dynamics.

The proposed model expands the understanding of chromosome congression beyond the conventional role of CENP-E as a motor protein. The study highlights a previously underappreciated inhibitory role of Aurora A in congression, which could have implications for cancer biology, particularly in the context of Aurora kinase overexpression. Given the importance of chromosome congression for accurate mitotic progression, these findings are highly relevant to cell biology and cancer research.

The following points should be addressed for improvement:

1) The working model is proposed heavily on data from experiments in which CENP-E was depleted or chemically inhibited. However, in unperturbed mitosis, CENP-E is usually active in either non-transformed cells or cancer cells. The authors should discuss the implication of their conclusion in normal mitosis progression.

The authors: We agree that unperturbed mitosis with active CENP-E warrants a more detailed discussion. We have now expanded on this point in the **Discussion** section:

„During unperturbed mitosis in healthy human cells, chromosomes typically remain at least 3 μm away from the centrosome and achieve rapid biorientation outside this region^{16,25,53}. We propose that in cells with active CENP-E most chromosomes rapidly establish end-on attachments and become bioriented close to the spindle surface by capturing microtubules extending from the opposite spindle half^{16,54}. Thus, while CENP-E supports microtubule stabilization at all kinetochores, as indicated by reduced microtubule density in kinetochore fibers upon its perturbation²⁸, its loss predominantly affects polar chromosomes without causing widespread detachment of those positioned farther from centrosomes.“

2) The author claimed the direct regulation of Aurora B by Aurora A. Direct biochemical assays demonstrating a physical or functional interaction between these two kinases would further substantiate the proposed model.

The authors: We agree with the reviewer that direct biochemical evidence of a physical or functional interaction between Aurora A and Aurora B would provide the most definitive support for our model. However, due to time constraints during the revision process, we prioritized a series of functional experiments. In these, Aurora A activity was selectively inhibited using three highly specific Aurora A inhibitors: MLN8054 (previously used in our study), TCS7010, and MLN8237 (Alisertib), in cells pre-treated with a CENP-E inhibitor.

We quantified the ratio of phosphorylated Aurora B (pAurB) at polar versus aligned kinetochores, as well as the number of polar chromosomes and spindle length per cell. Both MLN8054 and TCS7010 reduced the number of polar chromosomes by approximately threefold compared to CENP-E inhibition alone and significantly shortened spindle length, consistent with previous findings (doi: 10.1038/embor.2013.109). Importantly, pAurB levels at aligned kinetochores remained unchanged, indicating that these Aurora A inhibitors do not non-specifically impair Aurora B activity. These new data have been incorporated into the **Results** section.

“Acute inhibition of Aurora A significantly reduced Aurora B activity on polar kinetochores, bringing it to levels observed on aligned kinetochores in CENP-E–inhibited cells (Fig. 4a–c). To validate this finding, we tested two additional Aurora A–specific inhibitors: TCS7010³⁸, which also reduced pAurB asymmetry on polar versus aligned kinetochores similar to MLN8054, and Alisertib³⁹, which had no significant impact under the tested conditions (Fig. 4d, e). Consistent with these results, only MLN8054 and TCS7010 caused a ~2-fold decrease in the number of polar chromosomes and a significant reduction in spindle length⁴⁰ compared to CENP-E inhibition alone, without altering pAurB levels at aligned kinetochores (Fig. 4f, g; Extended Data Fig. 3f). These findings support a model in which Aurora A directly enhances Aurora B activity at polar kinetochores when CENP-E is inactive.”

While our data support a role for Aurora A in enhancing Aurora B function near spindle poles, the precise mechanism remains to be elucidated and will require future biochemical investigation. We now acknowledge this in the revised **Discussion** section.

“Without CENP-E, Aurora A overactivates Aurora B and its targets, such as KMN complex^{13,30,42,44} and the fibrous corona components^{48,49}, inhibiting end-on attachment stabilization and congression^{8,11,50} (Fig. 6). However, direct biochemical evidence for an Aurora A–Aurora B interaction has yet to be established.”

3) The discussion appropriately contextualizes the results, but alternative explanations for certain findings (such as potential indirect effects of Plk4 inhibition) should be considered.

The authors: We agree with the reviewer and have incorporated alternative interpretations for several of our findings, most notably those related to 1:0 spindles with only one centriolar spindle pole, into the **Discussion** section under “Limitations of the study” (see also a similar comment by Reviewer 3).

“In spindles where centrioles were removed by using the Plk4 inhibitor, potential indirect effects on chromosome congression cannot be fully excluded. For example, the asymmetric distribution of polar chromosomes may reflect the difference between the centriolar and acentriolar pole in their position and movement in addition to the difference in biorientation capacity. However, a similar trend was observed, albeit in a smaller number of cells, following spontaneous displacement of centrioles from spindle poles. Moreover, live imaging showed that chromosomes initially moved towards both poles, but they remained for a longer time at the centriolar pole (Video 1), supporting the robustness of our conclusions.”

4) The criteria for defining congression efficiency, as well as the statistical methods used for quantification, should be more explicitly described.

The authors: Those are now described in detail in the **Methods** section.

“The residence time of polar chromosomes on spindle poles, i.e., the efficiency of congression for each pole, was defined as the time from spindle bipolarization until the polar chromosome either entered the metaphase plate or the cell entered anaphase with an uncongressed polar chromosome.”

“In fixed cells, the number of polar chromosomes per single cell, as defined above, was used as a readout of congression efficiency.”

Statistical tests are now detailed in both the **Methods** section and the figure captions.

“The p values when comparing data from multiple classes that followed a normal distribution were obtained using the one-way ANOVA test followed by Two-sided Tukey's Honest Significant Difference (HSD) test (significance level was 5%).”

5) The discussion should address potential alternative mechanisms for the observed effects, particularly the role of additional spindle-associated proteins in congression regulation.

The authors: We agree with the reviewer and have included a discussion of potential roles of additional spindle-associated proteins in congression regulation in the **Discussion** section.

“While our study highlights the central role of CENP-E and Aurora kinases in chromosome congression, other spindle-associated proteins are likely involved. HURP and CLASP proteins, for instance, stabilize and regulate kinetochore microtubules near chromosomes^{56,57}, and the Ska complex enhances kinetochore–microtubule coupling under tension⁵⁸. At the spindle poles and in their vicinity, factors such as TPX2, pericentriolar proteins, kinesin-13 depolymerases, the crosslinker NuMA, and the Augmin complex are all intricately linked to Aurora A signaling³⁶. These pathways may act downstream of CENP-E and Aurora kinases to influence both congression efficiency and spatial bias, meriting further investigation.”

6) I would suggest the authors to provide additional experiments to address why the Hec1-9D mutant does not rescue CENP-E inhibition in the context of Hec1 depletion.

The authors: Although further investigation of this intriguing observation was beyond the scope of our revision timeline, we now clearly hypothesize two possible explanations for the phenotype. First, as discussed in detail in the accompanying manuscript, the fibrous corona, which is

overexpanded upon CENP-E inhibition but reduced during depletion in terms of CENP-E signal, may impede efficient microtubule capture, even when Hec1-9A is the only Hec1 variant present in the cell. Alternatively, additional phosphorylation sites on Hec1, such as S69, might play a critical role in regulating the initiation of congression under these conditions. Importantly, we performed an entirely new set of experiments using Hec1-9A with pre-synchronization, combined tubulin and CREST staining, and found that the results regarding CENP-E inhibition remained consistent (see also comment 3 by reviewer 1). We discuss these possibilities first in the **Results** section:

“Surprisingly, Hec1-9A overexpression did not rescue the congression defects caused by CENP-E inhibition in either synchronized or unsynchronized cells (Fig. 5b, c and e). This is likely due to hyperexpansion of the fibrous corona on polar kinetochores following CENP-E inhibition, which interferes with the stabilization of microtubule attachments¹¹. Consistent with this, the location and interkinetochore distance of polar kinetochores in cells overexpressing Hec1-9A under CENP-E inhibition (Fig. 5b) were similar to those under CENP-E inhibition alone (Fig. 4d). These findings suggest that constitutive Hec1 dephosphorylation is sufficient to drive chromosome congression in the absence of CENP-E and endogenous Hec1, but only when excessive fibrous corona expansion is avoided.”

We also address this in the **Discussion** section under the subsection titled ‘Limitations of the Study’:

“It remains unclear why CENP-E inhibition was not effectively rescued by overexpressing Hec1-9A in the context of Hec1 depletion, unlike with CENP-E depletion. One possibility is that the Hec1-S69 phosphorylation site, absent in the Hec1-9A mutant⁴⁴, is essential for bypassing the need for CENP-E under these conditions. Additionally, the extensive fibrous corona expansion might limit Hec1's engagement with microtubules in the cells with inhibited CENP-E⁴⁹. Further research is needed to clarify this.”

7) Where necessary, I would suggest the authors to include additional controls or validation to address the potential for non-specific binding of phospho-specific antibodies.

The authors: We thank the reviewer for raising this important point. To address it, we performed acute Aurora B inhibition in both control (DMSO-treated) and CENP-E-inhibited cells and stained for several phospho-specific antibodies used in this study: pAurB, pKnl1, and pHec1. The new data have been incorporated into the **Results** section:

“To confirm the specificity of phospho-antibodies as indicators of Aurora B activity, we performed acute Aurora B inhibition by 3 μ M ZM-4473 and assessed the localization and signal intensity of pAurB and pKnl1. Both signals were almost completely lost from kinetochores following inhibition, except for a small region near the centrioles, confirming their specificity as markers of Aurora B activity (Extended Data Fig. 3c). In contrast, the intensity and localization of pHec1 (pS55-Hec1)^{31,32} remained unchanged following acute Aurora B inhibition (Extended Data Fig. 3c), suggesting that S55 phosphorylation does not specifically reflect Aurora B activity, consistent with a recent report³³. This likely explains the uniform pHec1 levels at polar kinetochores regardless of the presence of centrioles on spindle poles and CENP-E activity (Extended Data Fig.

3d, e), which contrasts with the spatial variation and CENP-E dependence observed for pKn1 and pAurB (Fig. 3c–f).“

Reviewer #3 (Remarks to the Author):

In the work “Kinetochore-centrosome feedback linking CENP-E and Aurora kinases controls chromosome congression” the authors suggest a feedback loop involving Aurora Kinases and CENP-E. The novelty of the work lies in highlighting that centrosomes primarily inhibit congression initiation when CENP-E is inactive or absent. They have achieved this by removing centrioles using a PLK4 inhibitor, centrinone. This treatment allows chromosomes near acentriolar poles to initiate congression independently of CENP-E. Kinetochores near centriole bearing poles with high Aurora A activity further enhances Aurora B activity - this is thought to increase phosphorylation of microtubule-binding proteins at kinetochores and preventing stable microtubule attachments in the absence of CENP-E. This is rescued by inhibiting Aurora A as they find reduction in Aurora B activity, initiating congression without the CENP-E motor. Overall this is a good advance for the chromosome segregation field and enough detail is provided for the methods of the work to be reproduced.

Major comments

Figure 1: This is the strongest figure in the paper that shows the difficulty in congression near centrioles but not away – it would help if they could reproduce it in STLC-released cells to avoid any bias from end-point phenotypes (what if centriole bearing pole were dominant in the initial capture of chromosomes, then again one wouldn't see chromosomes at the centriole lacking pole). Why is figure 1C (Control) image blurrier compared to inhibitor and depleted conditions? Is this reflecting movements?

The authors: We thank the reviewer for raising this point. Spindles with 1:0 or 0:0 centrosome configurations naturally progress through a monopolar stage prior to bipolarization (see also doi: 10.1016/j.cub.2019.08.061). To determine whether the monopolar-to-bipolar transition influences the number of CENP-E-dependent kinetochores, we performed STLC washout before acute CENP-E inhibition, as described in the original manuscript (now Extended Data Fig. 2a). Our results showed that bipolarization from a monopolar spindle did not change the number of polar chromosomes compared to cells treated with the CENP-E inhibitor alone. We now discuss potential indirect effects of acentriolar spindle poles on congression efficiency in the **Discussion** section under the subsection titled ‘Limitations of the Study’:

“In spindles where centrioles were removed by using the Plk4 inhibitor, potential indirect effects on chromosome congression cannot be fully excluded. For example, the asymmetric distribution of polar chromosomes may reflect the difference between the centriolar and acentriolar pole in their position and movement in addition to the difference in biorientation capacity. However, a similar trend was observed, albeit in a smaller number of cells, following spontaneous displacement of centrioles from spindle poles. Moreover, live imaging showed that chromosomes initially moved towards both poles, but they remained for a longer time at the centriolar pole (Video 1), supporting the robustness of our conclusions.”

Figure 2 shows that Centriolar Aurora A activity is linked to increased activity of kinetochore Aurora B and its downstream KMN network targets without CENP-E activity. This is expected since Aurora A and B are known to have overlapping substrate recognition sites. However, this does not detract from the significance of their findings in this paper. Lack of centrioles allows the authors to elegantly establish the presence of excessive phosphoKNL1 in these conditions.

The authors: We agree with the reviewer and have now acknowledged the overlapping substrate recognition sites of the two kinases in the **Results** section:

“Given that polar kinetochores near centrosomes in the absence of CENP-E show elevated Aurora B phosphorylation, we hypothesized that Aurora A at spindle poles directly promotes Aurora B activity at kinetochores, consistent with their shared substrate recognition motifs^{33–35}.”

Figure 3 shows that Aurora-A directly upregulates Aurora-B using polar and nonpolar centromeres. This topic and conclusion have been reviewed before: PMC10040841. Can the authors help clarify how their work differs from past findings. If the difference is only the tool then Figures 2 and 3 can be easily combined and pAurB linked to the model. Have the authors tried other AuroraA inhibitors? MLN8054? or Aurora-A depletions. An alternate tool can help this part of the study to be stronger.

The authors: We have now included a discussion and citation of the study referenced by the reviewer in the **Results** section. While we acknowledge the conceptual possibility that Aurora kinases may share recognition sites and could potentially phosphorylate each other, as suggested in that study, we emphasize that this has not yet been experimentally validated nor has a clear physiological role for such cross-phosphorylation been established in the literature.

“Given that polar kinetochores near centrosomes in the absence of CENP-E show elevated Aurora B phosphorylation, we hypothesized that Aurora A at spindle poles directly promotes Aurora B activity at kinetochores, consistent with their shared substrate recognition motifs^{33–35}.”

To address the reviewer’s suggestion to test additional inhibitors, we employed three Aurora A–specific inhibitors: MLN8054 (previously used), TCS7010, and MLN8237 (Alisertib). We measured the ratio of phosphorylated Aurora B (pAurB) at polar versus aligned kinetochores, alongside quantifying the number of polar chromosomes and spindle length per cell. Consistent with our proposed model and live-cell imaging data (see accompanying manuscript), MLN8054 and TCS7010 reduced the number of polar chromosomes by approximately threefold and significantly shortened spindle length, consistent with prior findings (doi: 10.1038/embor.2013.109). Notably, pAurB levels on aligned kinetochores remained stable across treatments, indicating no nonspecific effect on global Aurora B activity. These new results have been incorporated into the **Results** section:

“Acute inhibition of Aurora A significantly reduced Aurora B activity on polar kinetochores, bringing it to levels observed on aligned kinetochores in CENP-E–inhibited cells (Fig. 4a–c). To validate this finding, we tested two additional Aurora A–specific inhibitors: TCS7010³⁸, which also reduced pAurB asymmetry on polar versus aligned kinetochores similar to MLN8054, and Alisertib³⁹, which had no significant impact under the tested conditions (Fig. 4d, e). Consistent

with these results, only MLN8054 and TCS7010 caused a ~2-fold decrease in the number of polar chromosomes and a significant reduction in spindle length⁴⁰ compared to CENP-E inhibition alone, without altering pAurB levels at aligned kinetochores (Fig. 4f, g; Extended Data Fig. 3f). These findings support a model in which Aurora A directly enhances Aurora B activity at polar kinetochores when CENP-E is inactive.”

Figure 4: This is probably the weakest figure of the study. Double depletions are hard to control. So it's ideal to assess the HEC1 and CENP-E protein levels independently in all conditions either with IF or IB. Frequently partially depleted kinetochores would result in congressed kinetochores while fully depleted ones would fail to align completely. Also, uncongressed chromosome measurements, as shown in Figure-4, must be combined with pole-to-pole axis. Especially since some of the treatments are also known to tumble the spindle position (for example the one presented in the leftmost panel has the entire metaphase rosette (and empty centre) visible compared to the HEC1 depletion. (middle panel). Since this is the main conclusion of the work, I would suggest measuring centromeric positions relative to spindle pole positions to confirm the outcomes.

The authors: We appreciate the reviewer's comment. To address these concerns, we performed a new set of experiments using HeLa cells expressing Hec1-9A-GFP. Cells were first synchronized in G1, then released and synchronized in G2 phase, during which endogenous Hec1 and/or CENP-E were depleted. This strategy ensured that cells did not enter mitosis lacking these proteins. After synchronization, cells were released into mitosis and incubated with a proteasome inhibitor for up to two hours, allowing metaphase arrest while minimizing confounding effects of prolonged mitotic arrest.

We analyzed four experimental conditions, staining all samples for tubulin and CREST:

1. Depletion of endogenous Hec1 and CENP-E
2. Depletion of endogenous Hec1 and CENP-E with expression of Hec1-9A-GFP
3. Depletion of endogenous Hec1 with expression of Hec1-9A-GFP
4. Depletion of endogenous Hec1 with expression of Hec1-9A-GFP combined with CENP-E inhibition.

All replicates were performed simultaneously using identical chemical treatments to ensure consistent depletion across conditions. While confirming depletion levels for each target would be ideal, this was not feasible due to channel limitations from tubulin, CREST, DNA labelling, and GFP-Hec1-9A expression. Verifying depletion would have required substituting one marker with antibodies against Hec1 or CENP-E, which was outside the scope of the current experiments.

To assess congression efficiency, we quantified both the maximum chromosome spread and the number of polar chromosomes per cell, the latter being independent of spindle pole-to-pole length. These new results have been incorporated into the **Results** section:

“Based on our finding that acute inhibition of Aurora kinases induces chromosome congression independently of CENP-E¹¹, we hypothesized that constitutive dephosphorylation of Hec1 might compensate for disruptions of CENP-E function after depletion of endogenous Hec1. To control for potential mitotic prolongation effects following combined Hec1 and CENP-E perturbations, we synchronized cells in G1 before depleting endogenous Hec1, CENP-E, or both, thereby ensuring depletion prior to mitotic entry. After release into G2 and mitosis, cells were arrested in metaphase using a proteasome inhibitor for a maximum of 2 hours (Fig. 5a, bottom).

We assessed chromosome congression efficiency by immunostaining cells for tubulin and centromeres under four conditions, all in the context of endogenous Hec1 depletion: and (1) CENP-E depletion, (2) Hec1-9A expression, (3) Hec1-9A expression with CENP-E depletion, (4) Hec1-9A expression with CENP-E inhibition. Expression of Hec1-9A in cells lacking endogenous Hec1 rescued major chromosome congression defects in both synchronized and unsynchronized cells (Fig. 5b, c, e), as previously reported^{43,44}. Remarkably, expression of Hec1-9A also significantly improved chromosome congression in cells depleted of both endogenous Hec1 and CENP-E (Fig. 5b). The improvement was reflected in both the maximum chromosome spread and the average number of polar chromosomes, which closely matched those observed in cells with intact CENP-E, in synchronized (Fig. 5b, e, Extended data Fig. 3j) as well as unsynchronized cells (Fig. 5c, Extended data Fig. 3i). In all conditions where Hec1-9A was overexpressed, aligned chromosomes had a large interkinetochore distance and showed end-on microtubule attachments (Fig. 5b). Surprisingly, Hec1-9A overexpression did not rescue the congression defects caused by CENP-E inhibition in either synchronized or unsynchronized cells (Fig. 5b, c and e). This is likely due to hyperexpansion of the fibrous corona on polar kinetochores following CENP-E inhibition, which interferes with the stabilization of microtubule attachments¹¹. Consistent with this, the location and interkinetochore distance of polar kinetochores in cells overexpressing Hec1-9A under CENP-E inhibition (Fig. 5b) were similar to those under CENP-E inhibition alone (Fig. 4d). These findings suggest that constitutive Hec1 dephosphorylation is sufficient to drive chromosome congression in the absence of CENP-E and endogenous Hec1, but only when excessive fibrous corona expansion is avoided.”

Minor comments

1. Do 1:1, 1:0 and 0:0 differ in their mitotic lengths or IK distances?

The authors: Mitotic duration varies across conditions, as shown in Fig. 1f, with the most pronounced differences observed between CENP-E-perturbed and unperturbed cells. Although inter-kinetochore (IK) distances were not measured in all conditions, data from chromosomes congressing from spindle poles containing either one centriole or no centrioles revealed a similar increase in IK distance during congression (now presented in Extended Data Fig. 2e).

2. In figure 3, P and A are clear? What happens to those on transit? Do they exclude them or assign them to either of the categories. They may have different behaviour- this is particularly important for pHEC1.

The authors: We thank the reviewer for this comment. We have now clearly defined the criteria for categorizing kinetochores as polar (P) or aligned (A) in the **Methods** section.

“In all experiments, polar kinetochores were defined as kinetochore pairs positioned closer to one of the spindle poles than to the center of the equatorial plane, with all others classified as aligned. In fixed-cell images, the polar category included both kinetochores near the pole and those in transit toward the metaphase plate. Kinetochore pairs with at least one kinetochore located 3 μm or less from the equatorial plane were assigned to the aligned group.”

Regarding the pHec1 data, since we have not confirmed the antibody’s specificity for Aurora B-mediated phosphorylation (see Reviewer 2, comment 7), we have moved this dataset to the Extended Data section. The new results are incorporated into the **Results** section:

“To confirm the specificity of phospho-antibodies as indicators of Aurora B activity, we performed acute Aurora B inhibition by 3 μM ZM-4473 and assessed the localization and signal intensity of pAurB and pKnl1. Both signals were almost completely lost from kinetochores following inhibition, except for a small region near the centrioles, confirming their specificity as markers of Aurora B activity (Extended Data Fig. 3c). In contrast, the intensity and localization of pHec1 (pS55-Hec1)^{31,32} remained unchanged following acute Aurora B inhibition (Extended Data Fig. 3c), suggesting that S55 phosphorylation does not specifically reflect Aurora B activity, consistent with a recent report³³. This likely explains the uniform pHec1 levels at polar kinetochores regardless of the presence of centrioles on spindle poles and CENP-E activity (Extended Data Fig. 3d, e), which contrasts with the spatial variation and CENP-E dependence observed for pKnl1 and pAurB (Fig. 3c–f).”

3. Please clarify the sentence “This effect, however, did not occur after CENP-E inhibition (Fig. 4C).”

The authors: We have now rewritten this section and clearly discussed the possible reasons why Hec1-9A did not improve chromosome alignment in cells with inhibited CENP-E (see major comments). We first discuss the possible reasons why Hec1-9A was unable to rescue congression in the presence of inhibited CENP-E in the **Results** section:

“Surprisingly, Hec1-9A overexpression did not rescue the congression defects caused by CENP-E inhibition in either synchronized or unsynchronized cells (Fig. 5b, c and e). This is likely due to hyperexpansion of the fibrous corona on polar kinetochores following CENP-E inhibition, which interferes with the stabilization of microtubule attachments¹¹. Consistent with this, the location and interkinetochore distance of polar kinetochores in cells overexpressing Hec1-9A under CENP-E inhibition (Fig. 5b) were similar to those under CENP-E inhibition alone (Fig. 4d). These findings suggest that constitutive Hec1 dephosphorylation is sufficient to drive chromosome congression in the absence of CENP-E and endogenous Hec1, but only when excessive fibrous corona expansion is avoided.”

We also address this issue in the **Discussion** section under the subsection titled “Limitations of the Study”:

“It remains unclear why CENP-E inhibition was not effectively rescued by overexpressing Hec1-9A in the context of Hec1 depletion, unlike with CENP-E depletion. One possibility is that the Hec1-S69 phosphorylation site, absent in the Hec1-9A mutant⁴⁴, is essential for bypassing the need for CENP-E under these conditions. Additionally, the extensive fibrous corona expansion might limit Hec1's engagement with microtubules in the cells with inhibited CENP-E⁴⁹. Further research is needed to clarify this.”

4. Minor points on discussion:

It is unclear why the authors do not conclude that kinetochore-centrosome feedback is between aurora kinases and the state of ‘attachment’. CENP-E loss is known to affect the status of attachment (lateral or not) which will broadly affect a variety of outer-kinetochore protein status and feedback loop.

The authors: We thank the reviewer for raising this important point. While we agree that CENP-E loss affects kinetochore-microtubule attachments and consequently alters outer kinetochore signaling, including Aurora B activity, our data, as well as previous studies (Eibes et al., 2023; Kim et al., 2010), demonstrate that Aurora A and B kinases also directly regulate CENP-E activity, establishing a bidirectional feedback loop. Therefore, the changes we observe are not merely downstream consequences of altered attachment status but reflect an interdependent regulatory circuit between kinase activities and motor protein function. We have now clarified this interplay more explicitly in the manuscript **Discussion** section:

“Our findings support a model in which kinetochore-centrosome feedback is not solely driven by the attachment status of microtubules to kinetochores but involves a bidirectional regulatory loop between Aurora kinases and CENP-E. While CENP-E promotes stable end-on attachments that suppress Aurora B activity, Aurora A and B themselves increase CENP-E activity and localization^{10,23}. This interdependence suggests that changes in kinetochore signaling reflect not just passive responses to the attachment state but active feedback between motor activity and kinase signaling near the spindle poles.”

5. Would this feedback loop be present or absent in oocytes which naturally lack centrioles and have large spindle pole to kinetochore distances?

The authors: We thank the reviewer for this insightful question. We have now addressed the relevance of this feedback loop in the context of oocytes by citing relevant literature (10.1073/pnas.94.17.9165; 10.1242/dev.078352). We propose that it is not simply the presence of centrioles, but rather the overall activity of Aurora A kinase, which is both present and essential for spindle assembly in oocytes (10.1083/jcb.201606077), that governs the feedback dynamics. While in somatic mitosis centriole presence correlates with elevated Aurora A activity, in oocytes, where canonical centrosomes are absent, Aurora A activity is sustained by alternative mechanisms. Further investigation of this feedback in oocytes could provide valuable insights into spindle assembly without centrioles. This discussion has now been incorporated into the **Discussion** section:

“Our findings suggest that it is not the presence of centrioles per se, but rather elevated Aurora A kinase activity that governs the feedback dynamics between spindle poles and kinetochores. In somatic cells, increased Aurora A activity is associated with canonical centrosomes, whereas oocytes maintain high Aurora A levels and form functional spindles without canonical centrosomes⁵⁹. This may explain why oocytes, but not acentriolar poles in somatic cells, require CENP-E for chromosome congression despite the absence of centrioles^{60,61}. We propose that spatial feedback mechanisms involving Aurora A may operate independently of centrioles, highlighting the need for further study of chromosome congression regulation in acentriolar contexts.”

6. Did they observe an increase in syntelic versus monotelic or 'behind the pole kinetochores' in their centrinone treatment study? Would polar chromosomes require centrioles to be captured normally?

The authors: We have now expanded the **Discussion** section to address spindle assembly in partially or fully acentriolar spindles in greater detail, acknowledging this as a potential caveat of the experiment:

“In spindles where centrioles were removed by using the Plk4 inhibitor, potential indirect effects on chromosome congression cannot be fully excluded. For example, the asymmetric distribution of polar chromosomes may reflect the difference between the centriolar and acentriolar pole in their position and movement in addition to the difference in biorientation capacity. However, a similar trend was observed, albeit in a smaller number of cells, following spontaneous displacement of centrioles from spindle poles. Moreover, live imaging showed that chromosomes initially moved towards both poles, but they remained for a longer time at the centriolar pole (Video 1), supporting the robustness of our conclusions.”

Regarding attachment types, we have not specifically examined the presence of syntelic or monotelic attachments. However, the limited STED images acquired from acentriolar cells (see Fig. 1) do not indicate a prevalence of these attachment types, consistent with observations of polar chromosomes in centriolar cells (see accompanying manuscript).

Rebuttal letter Vukušić&Tolić, NCOMMS-24-76220A

We thank the reviewers for highlighting parts that need to be clarified. Below, we provide detailed responses to each comment. All modifications to the manuscript are highlighted in blue for clarity.

REVIEWER COMMENTS

Reviewer #1 (Remarks to the Author):

Most of my concerns have been properly addressed. I think the manuscript is now suitable for publication.

Reviewer #2 (Remarks to the Author):

In the revised manuscript, the authors have addressed mostly of my concerns, through either carrying out additional experiments or extending the discussion. I therefore suggest that the revised manuscript is acceptable for publication.

The authors: We thank the reviewers 1 and 2 for assessing revised manuscript.

Reviewer #3 (Remarks to the Author):

In NCOMMS-24-76220A, the authors present evidence for a kinetochore-centrosome feedback linking CENP-E and Aurora kinases's role in chromosome congression. Their localised model for regulating chromosome congression is stronger with new experiments – including additional inhibitors, HEC1 and CENPE mutants.

The authors: We thank the reviewer 3 for evaluating revised manuscript.

Below are minor queries for clarity and accuracy.

Figure 2. It would help to clarify the conclusion on 'spatial bias': "These results support a model in which CENP-E-dependent congression of polar chromosomes is spatially biased toward centriolar spindle poles, both in non-transformed and transformed cells and across distinct modes of CENP-E perturbation (Fig. 2d)".

The authors: We have removed the word "spatially" from the sentence. The phrase "biased toward centriolar spindle poles" indeed inherently conveys a spatial direction, rendering the word "spatially" redundant.

Line 18: what does 'self-limiting' mean? Please describe in results or explain in discussion.

The authors: We thank the reviewer for raising this point. We have revised the sentence in Abstract to use "negative feedback mechanism" instead of "self-limiting feedback mechanism." Negative feedback is a very common and well-understood term in cell biology for mechanisms where the output of a process inhibits its own further production.

"We propose a negative feedback mechanism involving Aurora kinases and CENP-E that regulates the timing of chromosome movement by modulating kinetochore-microtubule attachments and fibrous corona expansion, with Aurora A activity gradient providing critical spatial cues for the network's function."

Line 117. This sentence requires clarification. "These findings suggest that centrioles specifically limit the initiation of polar chromosome congression when CENP-E activity is compromised." Clarifying this sentence is important because of a preceding sentence in 108 states "The acentriolar poles (labeled "0") in 1:0 spindles consistently had an average of around one polar chromosome, regardless of CENP-E activity".

The authors: We thank the reviewer for raising this point. We have modified the text to clarify that initial appearance of polar chromosomes at both pole types in 1:0 spindles are CENP-E-independent, whereas the prolonged persistence of these chromosomes is highly CENP-E-dependent only at centriolar centrosomes.

“As a proxy for congression efficiency, we quantified the number of polar chromosomes 5 minutes after spindle elongation, the residence time of polar chromosomes at the spindle poles, and the total duration of mitosis (Methods). Following CENP-E perturbations, spindles with a 1:1 or 2:2 centriole configuration had an average of about eight polar chromosomes per spindle, compared to fewer than one chromosome under control conditions in both spindle types (Fig. 1c and d; and Video 1). In 1:0 spindles after CENP-E perturbations, the centriolar poles (labeled "1") contained approximately four polar chromosomes, similar to their counterparts in 1:1 spindles (Fig. 1c, d; and Video 1). The acentriolar poles (labeled "0") in 1:0 spindles consistently had an average of around one polar chromosome (Fig. 1c and d; and Video 1). **In 1:0 spindles, the initial appearance of polar chromosomes at both the centriolar (labeled "1") and acentriolar (labeled "0") poles was independent of CENP-E activity (Fig. 1d). However, their subsequent persistence and efficient congression were critically dependent on CENP-E, particularly at centriolar poles. This is evidenced by the significantly longer retention of polar chromosomes at centriolar poles compared to acentriolar poles across all spindle types following CENP-E inhibition (Fig. 1e), resulting in a marked prolongation of mitosis (Extended Data Fig. 1e). Furthermore, completely acentriolar 0:0 spindles, which were rarely observed due to p53-dependent arrest in RPE-1 cells¹⁹, exhibited a low number of polar chromosomes under CENP-E inhibition, similar to the acentriolar poles in 1:0 spindles (Fig. 1c and f; and Video 1). These findings suggest that centrioles specifically limit the initiation of polar chromosome congression when CENP-E activity is compromised.**”

Line 223 Typo in inhibitor name: “inhibition by 3 μ M ZM-4473 and”

The authors: We have corrected this.

Line 337 “all in the context of endogenous Hec1 depletion” this work needs some evidence for endogenous Hec1 depletion. (Eg., cold treatment, endogenous Hec1 levels or an immunoblot to showcase loss of endogenous Hec1 as it’s a notoriously difficult protein to be fully depleted from the kinetochore).

The authors: We fully agree with the reviewer that Hec1 is notoriously difficult to deplete, a challenge we have also encountered in our experiments. To assess the level of Hec1 depletion, we performed immunofluorescence analysis of endogenous Hec1 at kinetochores following 3'-UTR Hec1 siRNA treatment in HeLa cells, as shown in Fig. 5d. The depletion was carried out using a previously optimized protocol (Etemad et al., 2015, *Nat. Commun.*). Nevertheless, we now acknowledge this well-known limitation of Hec1 depletion in the *Limitations of the Study* section.

“It remains unclear why CENP-E inhibition was not effectively rescued by overexpressing Hec1-9A in the context of Hec1 depletion, unlike with CENP-E depletion. One possibility is that the Hec1-S69 phosphorylation site, absent in the Hec1-9A mutant⁴⁴, is essential for bypassing the need for CENP-E under these conditions. Additionally, the extensive fibrous corona expansion might limit Hec1's engagement with microtubules in the cells with inhibited CENP-E⁴⁹. **Also, as Hec1 is known as notoriously difficult protein to be fully depleted from the kinetochore, due to its role as a core, stable kinetochore component³¹, some residual endogenous Hec1 might still be phosphorylated on the kinetochores.** Further research is needed to clarify this. Moreover, Aurora A might have a more direct impact on KMN network components, potentially through specific phosphorylation sites on Hec1, such as S69, which were not examined in this study^{32,50}.”

Line 375: Please double-check kinetochore numbers and include for all conditions if required “Numbers:571 kinetochore pairs from 91 cells (c), 327 cells (d) 203 cells (e), all from \geq 3 replicates”

The authors: We thank the reviewer for spotting this mistake in referencing numbers. This has been corrected and updated:

“Numbers: 360 cells (c), 65 cells (d) 214 cells (e), all from ≥ 3 replicates.”

Line 578: typo ‘final final’ “IC₅₀ value 3.4 nM) at a final final concentration of 100 nM,”

The authors: We have corrected this.

Multiple Aurora inhibitors are being named – some but not caused a phenotype. Please discuss the differences induced by these inhibitors (in results/discussion) as it would help future users. “Aurora A inhibitor Alisertib (MLN8237, MedChemExpress, IC₅₀ value 1.2 nM) at a final final concentration of 125 nM, was added 30 minutes before fixation. Aurora A inhibitor TCS7010 (MedChemExpress, IC₅₀ value 3.4 nM) at a final final concentration of 100 nM, was added 30 minutes before fixation. Aurora B inhibitor ZM-447439 (MedChemExpress, IC₅₀ value 130 nM) at a final concentration of 3 μ M, was added 15 minutes before fixation.”

The authors: We appreciate the reviewer's relevant comment regarding the varying effects observed with different Aurora A inhibitors. We have commented on possible reasons for the discrepancy between Aurora inhibitors in the *Methods* section.

“Spindle shortening was also observed in most CENP-E inhibited cells after acute addition of the Aurora A inhibitors MLN8054 (125 nM) and TCS7010 (100 nM), consistent with previous studies⁵⁰. Alisertib (125 nM) did not produce a comparable effect, likely due to the short treatment duration, limited to a maximum of 30 minutes, to accommodate the requirements of our assay. This brief incubation was necessary to prevent the re-congression of polar chromosomes, a phenomenon observed in untreated cells over time. Given that Alisertib is a well-characterized and potent Aurora A inhibitor⁵⁰, we suspect that a longer exposure or higher concentration might yield spindle shortening effects similar to those observed with MLN8054 and TCS7010.”

ZM 447439 can inhibit both Aurora A and B. Please comment or discuss.

The authors: We have incorporated this fact in the Methods section.

“Aurora A and B inhibitor ZM-447439 (MedChemExpress, IC₅₀ value 130 nM) at a final concentration of 3 μ M, was added 15 minutes before fixation.”

Rebuttal letter Vukušić&Tolić, NCOMMS-24-76220B

We thank the reviewer for highlighting Methods part that needs to be more detailed. Below, we provide detailed response to the comment. All modifications to the manuscript are highlighted in blue for clarity.

REVIEWERS' COMMENTS

Reviewer #3 (Remarks to the Author):

The authors have addressed my queries satisfactorily. Changes to results text and limitations text have made the work stronger.

As it's one of early lattice light sheet manuscript article, I would strongly recommend the authors to indicate details of light sheet thickness and length (not the imaging length). The details on laser illumination and objectives have all been added in the previous round of revision and these are useful, but light sheet details per se could be added for reproducibility.

The authors: We agree with the reviewer and have now added more details on lattice light sheet imaging, as requested.

Methods: “The Lattice Lightsheet 7 microscope system (Zeiss) was used for live cell imaging of hTERT-RPE1 cells expressing CENP-A-GFP and Centrin1-GFP in assay we termed the LLSM-based imaging assay. The system was equipped with an illumination objective lens 13.3×/0.4 NA (at a 30° angle to cover the glass) with a static phase element and a detection objective lens 44.83×/1.0 NA (at a 60° angle to cover the glass) with an Alvarez manipulator. Images were acquired using ZEN 2.7 (blue edition; Zeiss). The automatic immersion of water was applied from the motorized dispenser at an interval of 20 or 30 minutes. Right after sample mounting, three steps of the 'create immersion' auto immersion option were applied. The sample was illuminated with a 488-nm diode laser (power output 10 Mw) with laser power set to 1-2%. The detection module consisted of a Hamamatsu ORCA-Fusion sCMOS camera with exposure time set to 15-20 ms. The LBF 405/488/561/642 emission filter was used. During imaging, cells were kept at 37 °C and at 5% CO₂ in a Zeiss stage incubation chamber system (Zeiss). The imaging area's width in the x-dimension ranged from 1 to 1.5 mm, with a 0.4 μm interval size. The time between consecutive frames varied from 30 seconds to 1 minute, depending on the chosen imaging area width. The total imaging duration, set for 1 to 1.5 days, was occasionally interrupted by air bubbles, which caused a loss of intensity in part or all of the imaging area. When the entire area was affected by air bubbles, the image was cropped in ZEN software to reduce the final file size before further processing. Some movies, which were later processed for color-coding, were deskewed using ZEN 3.7 software with the "Linear Interpolation" and "Cover Glass Transformation" settings. The light sheet's length, also referred to as the field of view or illumination width, was 30 μm, while its thickness was set to 1000 nm. The parameters 'Focus sheet,' 'Focus Waist,' and 'Aberration Control' were manually fine-tuned before each imaging session, with ranges of -0.170 to -0.230, 50 to 60, and 170 to 185, respectively.”